# High-temperature $^{205}$Tl decay clarifies $^{205}$Pb dating in early Solar System

Guy Leckenby[1,2✉], Ragandeep Singh Sidhu[3,4,5], Rui Jiu Chen[3,5,6✉], Riccardo Mancino[3,7,36], Balázs Szányi[8,9,10], Mei Bai[3], Umberto Battino[11,12], Klaus Blaum[5], Carsten Brandau[3,13], Sergio Cristallo[14,15], Timo Dickel[3,16], Iris Dillmann[1,17], Dmytro Dmytriiev[3,18], Thomas Faestermann[19], Oliver Forstner[3,20], Bernhard Franczak[3], Hans Geissel[3,16], Roman Gernhäuser[19], Jan Glorius[3], Chris Griffin[1], Alexandre Gumberidze[3], Emma Haettner[3], Pierre-Michel Hillenbrand[3,13], Amanda Karakas[21,22,23], Tejpreet Kaur[24], Wolfram Korten[25], Christophor Kozhuharov[3], Natalia Kuzminchuk[3], Karlheinz Langanke[3], Sergey Litvinov[3], Yuri A. Litvinov[3,26✉], Maria Lugaro[9,10,21,27✉], Gabriel Martínez-Pinedo[3,7,26], Esther Menz[3], Bradley Meyer[28], Tino Morgenroth[3], Thomas Neff[3], Chiara Nociforo[3], Nikolaos Petridis[3], Marco Pignatari[9,10,11], Ulrich Popp[3], Sivaji Purushothaman[3], René Reifarth[29,30], Shahab Sanjari[3,31], Christoph Scheidenberger[3,16,32], Uwe Spillmann[3], Markus Steck[3], Thomas Stöhlker[3,20], Yoshiki K. Tanaka[33], Martino Trassinelli[34], Sergiy Trotsenko[3], László Varga[3,37], Diego Vescovi[14,15,29], Meng Wang[6], Helmut Weick[3], Andrés Yagüe Lopéz[30], Takayuki Yamaguchi[35], Yuhu Zhang[6] & Jianwei Zhao[3]

Radioactive nuclei with lifetimes on the order of millions of years can reveal the formation history of the Sun and active nucleosynthesis occurring at the time and place of its birth[1,2]. Among such nuclei whose decay signatures are found in the oldest meteorites, $^{205}$Pb is a powerful example, as it is produced exclusively by slow neutron captures (the $s$ process), with most being synthesized in asymptotic giant branch (AGB) stars[3–5]. However, making accurate abundance predictions for $^{205}$Pb has so far been impossible because the weak decay rates of $^{205}$Pb and $^{205}$Tl are very uncertain at stellar temperatures[6,7]. To constrain these decay rates, we measured for the first time the bound-state $\beta^-$ decay of fully ionized $^{205}$Tl$^{81+}$, an exotic decay mode that only occurs in highly charged ions. The measured half-life is 4.7 times longer than the previous theoretical estimate[8] and our 10% experimental uncertainty has eliminated the main nuclear-physics limitation. With new, experimentally backed decay rates, we used AGB stellar models to calculate $^{205}$Pb yields. Propagating those yields with basic galactic chemical evolution (GCE) and comparing with the $^{205}$Pb/$^{204}$Pb ratio from meteorites[9–11], we determined the isolation time of solar material inside its parent molecular cloud. We find positive isolation times that are consistent with the other $s$-process short-lived radioactive nuclei found in the early Solar System. Our results reaffirm the site of the Sun's birth as a long-lived, giant molecular cloud and support the use of the $^{205}$Pb–$^{205}$Tl decay system as a chronometer in the early Solar System.

The presence of radioactive nuclei with astrophysically short half-lives—roughly between 1 and 100 Myr—at the time of the formation of the first solids in the early Solar System is well documented from the laboratory analysis of meteorites and the incorporated mineral inclusions[1,2]. As the Sun is roughly 4.6 billion years old, these nuclei have now fully decayed. However, their live abundances in the early Solar System—in the form of ratios to a stable isotope of the same element—can be derived from measurable excesses in the abundances of their decay-daughter nuclei (prescription in Methods). This abundance snapshot provides us information on the nucleosynthetic events before the formation of the Solar System, as well as details on the chronology of early Solar System evolution[12–14]. For example, the decay time

required for the abundance ratio predicted in the interstellar medium (ISM) to reach the ratio measured in meteorites can represent the isolation time of Solar System material inside its parent molecular cloud before the birth of the Sun[1,15,16].

Among the 18 measurable short-lived radionuclides produced in stellar environments, four are produced by slow neutron captures (the $s$ process) in AGB stars: $^{107}$Pd ($t_{1/2}$ = 6.5(3) Myr), $^{135}$Cs (1.33(19) Myr), $^{182}$Hf (8.90(9) Myr) and $^{205}$Pb (17.0(9) Myr)[17]. Although the first three (and their stable reference isotopes, $^{108}$Pd, $^{133}$Cs and $^{180}$Hf) can also be produced by rapid neutron captures (the $r$ process), $^{205}$Pb and its stable reference isotope $^{204}$Pb are shielded from the $\beta^-$-decay chains of $r$-process production by stable $^{204}$Hg and $^{205}$Tl (see Fig. 1a). From

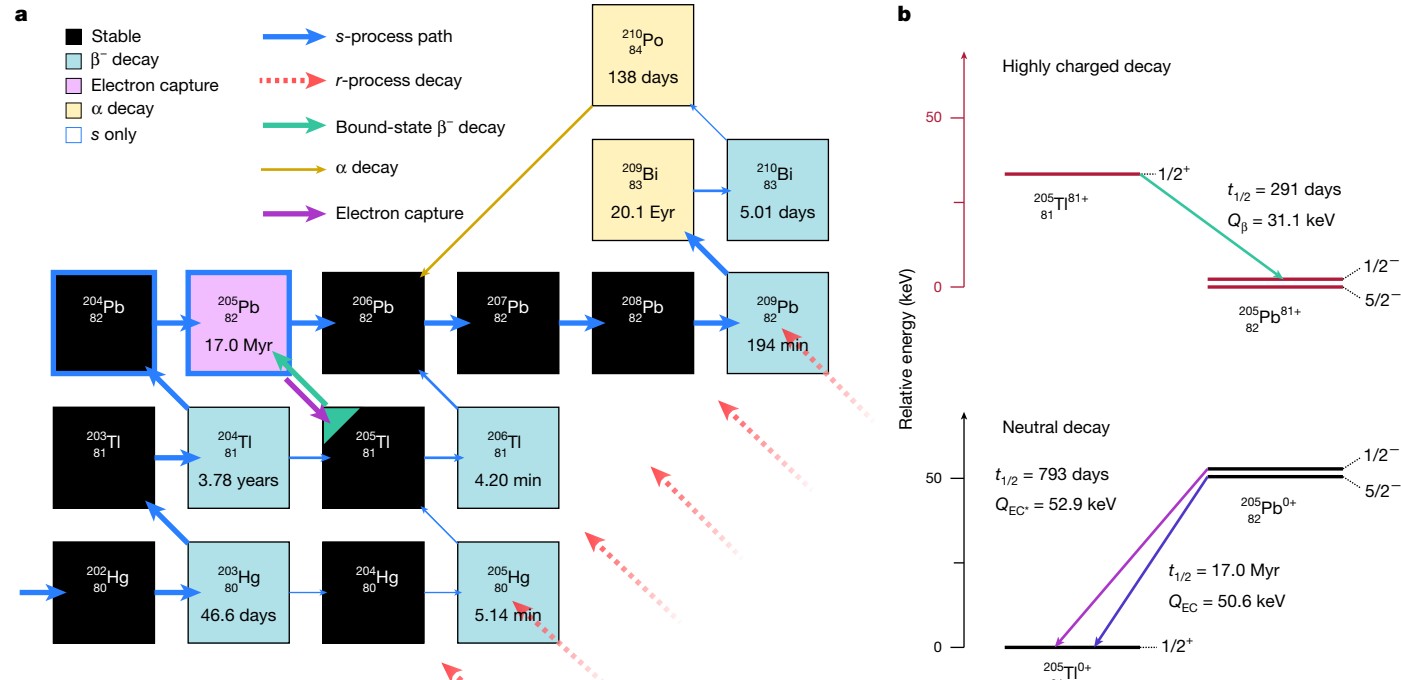

**Fig. 1 | s-process reaction path around Tl and Pb. a**, $^{205}$Pb is situated at the end of the s-process path, represented by the sequence of n captures and β decays shown by the blue arrows. Galactic production of $^{205}$Pb is exclusive to the s process, which makes it unique among short-lived radionuclides. **b**, Low-lying nuclear structure of the $^{205}$Pb–$^{205}$Tl system. Decay of neutral $^{205}$Pb proceeds by electron capture from the ground state (purple) or from the excited state (magenta) if thermally populated. In fully ionized conditions (shown in red), bound-state β$^-$ decay of $^{205}$Tl$^{81+}$ occurs (green) by decay to the 2.3-keV excited state. All half-lives are partial half-lives for that specific transition, but it should be noted that the 2.3-keV excited state is never populated in a neutral atom, and full temperature-dependent and density-dependent rates need to be used.

predictions of the evolution of the galactic abundances of $^{107}$Pd, $^{135}$Cs and $^{182}$Hf, self-consistent isolation times in the range 9–26 Myr have been obtained[18]. This relatively high range confirms that the Sun was born in a giant molecular cloud, such as Scorpius–Centaurus OB2 (ref. 19) and the Orion molecular cloud complex[20], with a long lifetime and nursing many stellar generations. This conclusion relies on the proposed scenario that the last r-process event to have contributed the r-process short-lived radionuclides ($^{129}$I, $^{244}$Pu and $^{247}$Cm) to the presolar material occurred 100–200 Myr before the formation of the first solids[21,22]. This longer time period would greatly suppress the r-process contribution for the s-process short-lived radionuclides, $^{107}$Pd, $^{135}$Cs and $^{182}$Hf. $^{205}$Pb represents a critical test for this scenario, given that it does not have any r-process contribution.

A notable issue that has prevented the use of $^{205}$Pb as a cosmochronometer is the temperature dependence of its decay rate, as noted by Blake and Schramm[6] soon after introducing the chronometer in 1973 (ref. 23). The bound-electron capture from the ground state of $^{205}$Pb (spin and parity 5/2$^-$) to the ground state of $^{205}$Tl (1/2$^+$) is strongly suppressed owing to the very different structures of the two states, resulting in the long half-life of 17 Myr at terrestrial temperatures. However, $^{205}$Pb has a low-lying, first excited state at 2.3 keV with a spin and parity of 1/2$^-$, shown in Fig. 1b. No suppression resulting from the nuclear structure occurs for decay from this 1/2$^-$ excited state, resulting in a decay rate that is 5–6 orders of magnitude faster than from the 5/2$^-$ ground state. Blake and Schramm[6] realized that the stellar temperatures achieved in AGB stars will thermally populate this spin-1/2 excited state, causing most synthesized $^{205}$Pb to decay before it can be ejected from the star into the ISM.

Yokoi et al.[7] countered in 1985 that, at s-process temperatures, the bound-state β$^-$ decay of $^{205}$Tl could produce enough $^{205}$Pb to compete with the enhanced stellar decay rate. Although the β$^-$ decay of $^{205}$Tl to the continuum is not energetically allowed, β$^-$ decay to a bound state is possible if the β electron is created directly in the K shell of the daughter nucleus[24,25]. The binding energy of this bound state becomes available for the decay, making the Q value positive with $Q_{\beta_b,K} = 31.1(5)$ keV (derivation in Methods) (refs. 26–28). However, the K shell is only unoccupied in 80+ and 81+ charged ions of $^{205}$Tl, so if the nucleus exists in these high charge states, then bound-state β$^-$ decay becomes possible and $^{205}$Tl$^{80/81+}$ decays overwhelmingly to the 2.3-keV excited state of $^{205}$Pb$^{80/81+}$ (see Fig. 1b).

The temperatures required to populate such high charge states of $^{205}$Tl in the stellar plasma are reached at the s-process site in AGB stars. In fact, the layer between the H-burning and He-burning shells, in which the $^{13}$C(α, n)$^{16}$O and $^{22}$Ne(α, n)$^{25}$Mg neutron sources are activated, ranges in temperature from 90 to 370 MK (7.8–31.9 keV)[3]. The $^{205}$Tl bound-state β$^-$-decay rate is expected to compete with the $^{205}$Pb excited-state decay rate over exactly this temperature range, and when neutron captures are included, the specific temperature and density trajectory throughout this intershell region will determine which element dominates. This dynamic, temperature-dependent decay pairing has been modelled using increasingly complex stellar physics[7,29], but accurate yield predictions have been hampered by the fact that both decay rates are theoretically uncertain by orders of magnitude.

The weak decay rates of both $^{205}$Pb and $^{205}$Tl under stellar conditions are determined by the same transition between the spin-1/2 states. Measuring the half-life in either direction provides us with the nuclear matrix element of the transition, which will allow us to calculate precise astrophysical decay rates. The nuclear matrix element quantifies the wavefunction overlap between initial and final states in a decay, describing how similar the two nuclear configurations are and, therefore, how easy it is for the nucleus to decay. In the laboratory, the $^{205}$Pb excited state decays to the ground state through an internal conversion with a half-life of 24 μs (ref. 30). Thus, measuring the bound-state β$^-$ decay of $^{205}$Tl$^{81+}$ is the only way to directly measure the weak nuclear matrix element between the two states.

## 205Tl experiment

The measurement of the bound-state β⁻ decay of $^{205}\text{Tl}^{81+}$ was proposed in the 1980s and was one of the motivational cases[31] for the construction of the experimental storage ring (ESR)[32] at GSI Helmholtzzentrum für Schwerionenforschung in Darmstadt. However, this experiment was extremely challenging and, in spite of being on the highest priority list at GSI, could only be accomplished now, 40 years after its inception. To measure the bound-state β⁻ decay, $^{205}\text{Tl}$ nuclei needed to be fully stripped of electrons and stored for several hours to accumulate enough decay statistics. Creating a thallium ion beam is problematic as its vapours are highly toxic, resulting in prohibitive safety considerations for most ion sources. To circumvent this, we created $^{205}\text{Tl}^{81+}$ ions by means of a nuclear reaction known as projectile fragmentation, in which nucleons are removed from the projectile nuclei by a collision with a light target. We used a $^{206}\text{Pb}$ primary beam accelerated to 678 MeV per nucleon (81.6% of the speed of light) using the entire accelerator chain at GSI[33] and then collided the beam with a $^{9}\text{Be}$ target. The resulting fragmentation was dominated by the knockout of a few nucleons, with $^{205}\text{Tl}$ produced by one-proton knockout. The 81+ charge state was isolated by the fragment separator (FRS) using magnetic dipole separation before and after energy-loss selection as the beam passed through an energy degrader ($B\rho_1 - \Delta E - B\rho_2$) (ref. 34). The layout of GSI and the accelerator systems used to create, purify and store the $^{205}\text{Tl}^{81+}$ ions are shown in Fig. 2a.

Heavy-ion storage rings are uniquely capable of storing millions of fully stripped heavy ions for several hours, during which the ions are steered into cyclic trajectories and left to revolve. The combination of the FRS and the ESR is at present the only facility that can provide stored, fully stripped $^{205}\text{Tl}^{81+}$ ions. These ions were injected into the storage ring and about 99.9% of critical $^{205}\text{Pb}^{81+}$ contaminants co-produced during projectile fragmentation was blocked using slits at the exit of the FRS. To achieve more than $10^6$ orbiting $^{205}\text{Tl}^{81+}$ ions, up to 200 injections were accumulated in the storage ring. The ions were then continuously cooled by a beam of monoenergetic electrons and stored for up to 10 h, allowing decays to accumulate. These long beam lifetimes were achieved by operating the entire ring at ultrahigh-vacuum conditions of $<10^{-11}$ mbar. The stacking and storage of $^{205}\text{Tl}^{81+}$ ions is shown in Fig. 2b.

During the storage time, the parent $^{205}\text{Tl}^{81+}$ ions decayed by bound-state β⁻ decay to $^{205}\text{Pb}^{81+}$ daughter ions. Because the β electron was created in a bound state of the $^{205}\text{Pb}$ nucleus, the daughter ions had the same charge state as the $^{205}\text{Tl}^{81+}$ parent ions, so the mass-to-charge ratio changed only by the Q value of the decay, that is, only 31.1(5) keV. With such a small mass difference, the two beams of parent and daughter ions were mixed together and thus indistinguishable. Hence, to count the number of decayed ions, an argon gas-jet target was turned on at the end of the storage period that interacted with the entire beam, which stripped off the bound electron from the $^{205}\text{Pb}^{81+}$ daughter ions, leaving them in the 82+ charge state.

The growth of the $^{205}\text{Pb}/^{205}\text{Tl}$ ratio with storage time is determined by the bound-state β⁻ decay of $^{205}\text{Tl}^{81+}$ ions, as no other decay modes were possible. Because the half-life is much longer than our storage times, the observed growth is well approximated by the linear relation:

$$\frac{N_{\text{Pb}}(t_s)}{N_{\text{Tl}}(t_s)} = \frac{\lambda_{\beta_b}}{\gamma} t_s \left[ 1 + \frac{1}{2}(\lambda_{\text{Tl}}^{\text{loss}} - \lambda_{\text{Pb}}^{\text{loss}}) t_s \right] + \frac{N_{\text{Pb}}(0)}{N_{\text{Tl}}(0)} \exp[(\lambda_{\text{Tl}}^{\text{loss}} - \lambda_{\text{Pb}}^{\text{loss}}) t_s],$$

(1)

in which $N_X$ is the number of $^{205}\text{Pb}$ or $^{205}\text{Tl}$ ions, $t_s$ is the storage time, $\lambda_{\beta_b}$ is the bound-state β⁻-decay rate of $^{205}\text{Tl}^{81+}$ and $\gamma = 1.429(1)$ is the Lorentz factor for conversion into the laboratory frame. The effects of beam losses owing to electron recombination in the storage ring, given by

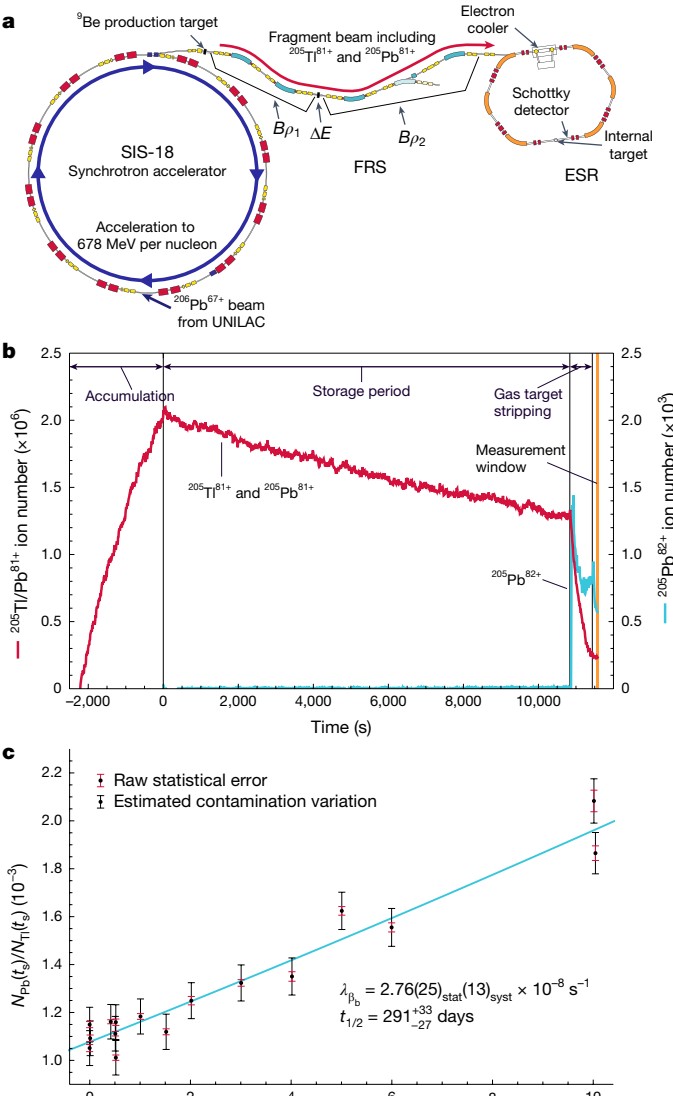

**Fig. 2 | $^{205}\text{Tl}$ experiment setup and results. a**, A $^{206}\text{Pb}^{67+}$ beam was accelerated by the SIS-18 synchrotron and projectile fragmentation produced $^{205}\text{Tl}^{81+}$ ions that were selected by the FRS and stored in the ESR. Copyright, GSI/FAIR. **b**, Ion intensity monitoring shows the accumulation and storage of $^{205}\text{Tl}^{81+}$ ions, as well as the revelation of $^{205}\text{Pb}^{82+}$ daughter ions once the gas target is turned on. Note that the $^{205}\text{Pb}^{82+}$ intensity is scaled by $10^3$ so that it is visible. **c**, The observed ratio of $^{205}\text{Pb}^{81+}$ to $^{205}\text{Tl}^{81+}$ ions for 16 storage runs is shown, alongside the best fit of equation (1) (blue line). Error bars are 1σ Gaussian.

$\lambda_X^{\text{loss}}$, must be included when solving the differential equation, although only the difference between these loss rates affects the ratio.

The best fit of equation (1) to the 16 storage runs is shown in Fig. 2c and yielded an observed decay rate of $\lambda_{\beta_b} = 2.76(25)_{\text{stat}}(13)_{\text{syst}} \times 10^{-8} \text{ s}^{-1}$. The uncertainties from statistics and corrections are given at 1σ and were propagated with Monte Carlo resampling ($10^6$ samples) to handle the correlation of some corrections between storage times. The statistical uncertainty is dominated by the variation in the $^{205}\text{Pb}^{81+}$ contamination from the fragmentation reaction, whose extreme kinematic tails extended beyond the collimating slits and made it into the storage ring (details in Methods).

The measured decay rate is equivalent to a half-life of $291^{+33}_{-27}$ days, or $\log(ft) = 5.91(5)$. The quantity $ft$ is often used to describe the magnitude of the transition, as it is inversely proportional to the square of the nuclear matrix element, thus removing the phase-space dependence of the decay rate. Theoretical predictions for the $\log(ft)$ of the

bound-state β⁻ decay of ²⁰⁵Tl⁸¹⁺ have been made from systematic extrapolation of nearby nuclei, yielding a range of values: log($ft$) = 5.1–5.8 (refs. 35–40). Our measured value is larger than that predicted from systematic extrapolation, highlighting the importance of our experimental result and the improved uncertainty of the ²⁰⁵Pb–²⁰⁵Tl decay scheme. We note that ²⁰⁵Tl can be used for detecting solar pp neutrinos[41]. The half-life measured here can be used to constrain the neutrino capture cross-section, which will be reported elsewhere[42].

## New weak decay rates

The β-decay rates of ²⁰⁵Pb and ²⁰⁵Tl in stellar plasma that have been used by most astrophysical models of ²⁰⁵Pb production were calculated by Takahashi and Yokoi[8]. Their rates were based on an extrapolated log($ft$) = 5.4 and a (now outdated) $Q_{\beta_b}$ = 40.3 keV, yielding a half-life of 58 days. Our experimental half-life is 4.7 times longer, resulting in reduced decay rates for both the excited-state decay of ²⁰⁵Pb and the bound-state β⁻ decay of ²⁰⁵Tl⁸⁰/⁸¹⁺.

To calculate revised temperature-dependent and density-dependent astrophysical decay rates for ²⁰⁵Pb and ²⁰⁵Tl, we followed the prescription for handling β-decay rates of highly ionized heavy atoms outlined in ref. 43. To calculate the decay rate at a given temperature and density, the distribution of ²⁰⁵Pb and ²⁰⁵Tl ions in the plasma is computed using the Saha equation, accounting for the Coulomb interaction of the ion with free electrons that reduces the ionization potential. The population of excited nuclear states is assumed to follow Boltzmann statistics. The total decay rate for each isotope is the weighted sum of decays from the nuclear excitation and ionization populations. Our rates are based on a shell-model calculation of all the relevant matrix elements calibrated to the measured rates and hence accounts for the full phase-space dependence of forbidden decays (see Methods).

Our new recommended rates are plotted in Fig. 3a and include both continuum and bound contributions. The thermal population of the 2.3-keV excited state substantially enhances the bound-electron-capture rate of ²⁰⁵Pb from temperatures of 2 MK to roughly 50 MK. Above 50 MK, the rate decreases owing to increasing ionization, which reduces the available bound electrons to be captured. The magnitude of this decrease depends on the density, as increased density both suppresses ionization and increases the rate of continuum electron capture. The opening of the bound-state β⁻-decay channel is simply proportional to the number of ions in the 80+/81+ charge states and occurs at higher temperatures for higher densities owing to the suppression of ionization.

In Fig. 3b, our rates are compared with the rates used at present in stellar models: those tabulated by Takahashi and Yokoi[8] (on which the extrapolated rates used in the FRUITY models[44] are based) and those recommended in the NETGEN library[45] (taken from ref. 46). Note that the diverging behaviour at low temperatures between FRUITY and NETGEN is because of the different space of linear interpolation to the terrestrial value (log versus linear, respectively). Relative to these previous rates, our new rates represent a considerable step forward. As well as being based on the new, longer, experimental half-life of ²⁰⁵Tl⁸¹⁺, we have also implemented the $Q$ value from the latest Atomic Mass Evaluation[26], based primarily on the measurements in ref. 47, and expanded the range and resolution of the temperature and density grid with modern computing power to $T$ = 0.5–500 MK and $n_e$ = 10²¹–10²⁸ cm⁻³.

Alongside the new weak rates, we have also revised the neutron-capture cross-sections for nine crucial isotopes, namely, ²⁰²⁻²⁰⁴Hg, ²⁰³⁻²⁰⁵Tl and ²⁰⁴⁻²⁰⁶Pb, to supplement the Karlsruhe Astrophysical Database of Nucleosynthesis in Stars (KADoNiS)[48] that is used by stellar models. These new recommendations, which include new experimental constraints and revised semiempirical estimates (see Methods), ensured that our predicted ²⁰⁵Pb yields are as up to date as possible.

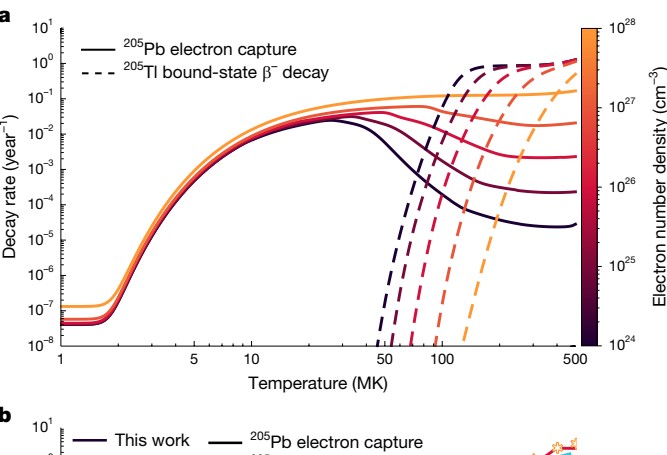

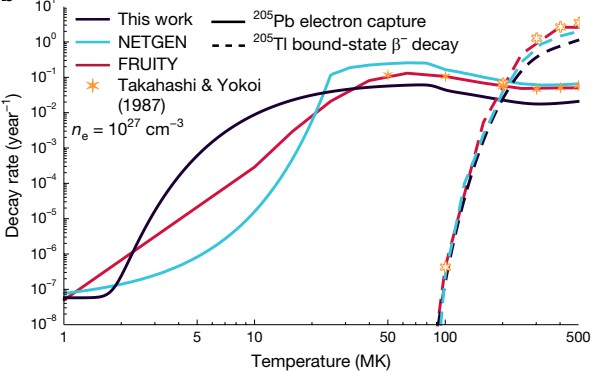

**Fig. 3 | Temperature-dependent and density-dependent decay rates for ²⁰⁵Pb and ²⁰⁵Tl. a**, Our new weak decay rates across astrophysically relevant conditions. The ²⁰⁵Pb rate increases with thermal population of the 2.3-keV excited state, whereas the bound-state β⁻-decay channel opens as K-shell ionized states are populated ($T$ > 50 MK). **b**, Our new rates compared with the original rates published in ref. 8, their interpolation as used in the FRUITY models[44] and the rates recommended in the NETGEN library[45], for $n_e$ = 10²⁷ cm⁻³.

## AGB stellar models

The $s$ process in AGB stars is driven by two neutron sources[3]. Recurrent episodes of partial mixing at the interface of the H-rich convective envelope and He-rich shell cause ¹²C and protons to mix and produce ¹³C, a neutron source, by means of ¹³C(α, $n$)¹⁶O for temperatures above 90 MK. This reaction produces a large number of neutrons (with relatively low neutron densities of about 10⁷ cm⁻³) during the long (approximately 10⁴ years) intervals of H-shell burning, between periodic thermonuclear He-burning runaways of the He-rich shell (thermal pulses). The contribution of the ¹³C neutron source to the production of ²⁰⁵Pb is of little importance however, because electron densities (on the order of 10²⁷–10²⁸ cm⁻³) and temperatures (on the order of 100 MK) are such that ²⁰⁵Pb decay is activated over the long intervals between thermal pulses, whereas ²⁰⁵Tl decay is not. The bulk of ²⁰⁵Pb is instead produced during the thermal pulses, for which the temperature can reach above 300 MK, resulting in a marginal neutron flux produced by the ²²Ne(α, $n$)²⁵Mg reaction (with relatively high neutron densities up to around 10¹² cm⁻³) and lasting several years. During this neutron flux, the complex interplay between decays and neutron captures on ²⁰⁴,²⁰⁵Pb and ²⁰³,²⁰⁴,²⁰⁵Tl results in a ²⁰⁵Pb/²⁰⁴Pb ratio on the order of unity. The final abundances available to be dredged up to the stellar surface, by means of the recurrent third dredge-up episodes that occur after each thermal pulse, are set during the phase between the end of each thermal pulse and the start of the following dredge-up. In this phase, no neutrons are available but ²⁰⁵Tl and ²⁰⁵Pb continue to decay according to the local temperature and electron density. Once carried to the convective envelope, the ²⁰⁵Pb abundance is preserved and ejected in the ISM by means of stellar winds.

To calculate quantitatively the total amount of $^{205}$Pb ejected by AGB winds (that is, the stellar yield), we have implemented the new weak rates into AGB models of solar metallicity in the mass range 2.0–4.5 solar masses using the Monash stellar evolution and nucleosynthesis tools[49]. We found that the yield of $^{205}$Pb increases by a factor of roughly 3.5–7.0 (the exact value depending on the stellar mass and metallicity) compared with using the rates tabulated in the NETGEN database[45]. As well as the Monash models, we also computed FUNS stellar evolution models[50] (used to produce the FRUITY database[44]) and NuGrid models[51]. Each model had a different response to our new rates, both because each model used a slightly different original grid for the decay rates (see Fig. 3b) and because each model operates at a different temperature during the crucial interval between the thermal pulse and the third dredge-up owing to a variety of astrophysical considerations (described in Methods). The revised neutron cross-sections changed the yields by less than 10%.

## $^{205}$Pb in the early Solar System

The AGB stars that we modelled (2.0–4.5 solar masses) are those responsible for the production of Pb $s$-process abundances in the Galaxy[49,52]. To simulate production from a stellar population, we weighted our $^{205}$Pb yields by the initial stellar mass using Salpeter's mass function (a basic power-law description of the mass distribution of a stellar population) and derived an average $^{205}$Pb/$^{204}$Pb production ratio of $P = 0.167$. From this production ratio, the $^{205}$Pb/$^{204}$Pb ratio in the ISM, at the galactic age $T_{Gal} = 8.4$ Gyr corresponding to the birth of the Sun[18], is represented by the steady-state abundance reached by the balance between production events and radioactive decay. This ratio is given by:

$$\left(\frac{^{205}\text{Pb}}{^{204}\text{Pb}}\right)_{ISM} = K \times P \times \frac{\tau_{205}}{T_{Gal}}, \quad (2)$$

in which $\tau_{205} = 24.5$ Myr is the mean lifetime of $^{205}$Pb and the parameter $K = 2.3$ represents the effect of various features of galactic evolution[53] (see Methods for an analysis of the uncertainties of the value of $K$). The derived ISM $^{205}$Pb/$^{204}$Pb ratio is $(1.10^{+0.30}_{-0.27}) \times 10^{-3}$, for which the statistical uncertainties are given at $1\sigma$ and derived from a previous Monte Carlo statistical analysis of the fact that the production rate from stars is a stochastic and unevenly distributed process[54].

The time required for the ISM ratio to decay to the value measured in meteorites is the interval between the isolation of the Sun's parent molecular cloud from further enrichment to the formation of the first solids in the early Solar System. For the meteoritic value, we used both $1.8(12) \times 10^{-3}$ as recommended in ref. 9, which covers the full range of $2\sigma$ uncertainties when combining the results by two separate experiments on two types of meteorite (primitive carbonaceous chondrites and iron meteorites)[10,11], and the original value of $1.0(4) \times 10^{-3}$ from the carbonaceous chondrites. The derived time intervals span physical (positive) values for 25% and 78% of the probability density for the full range and carbonaceous chondrites value, respectively, as shown in Fig. 4, which were impossible to obtain with the Monash calculations using the previous decay rates from the NETGEN compilation. The positive isolation times for $^{205}$Pb are in agreement with those derived from $^{107}$Pd and $^{182}$Hf, especially when considering the values in the lower range of the $^{205}$Pb/$^{204}$Pb ratio in the early Solar System provided by the original analysis of carbonaceous chondrite meteorites[10]. The recommended mean value can also provide a consistent solution when considering the uncertainties related to $K$.

With the newly measured value of bound-state $\beta^-$-decay half-life for $^{205}$Tl$^{81+}$ and our improved astrophysical rates, a self-consistent scenario for the $s$-process short-lived radionuclides in the early Solar System is found, which was not possible before. From this initial analysis, the proposed scenario of the Sun forming inside a giant molecular cloud,

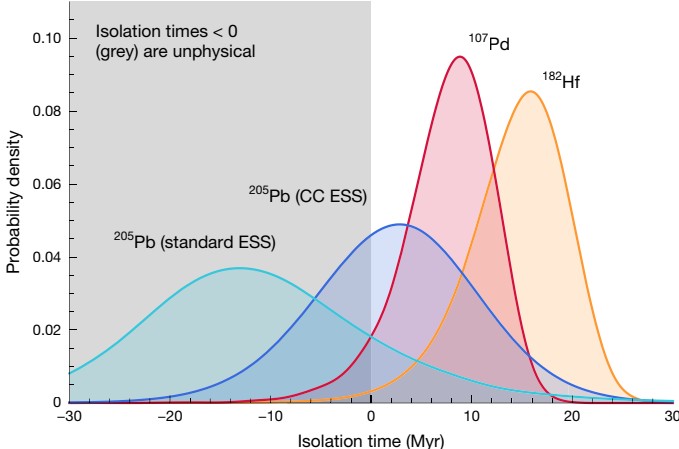

**Fig. 4 | Probability density functions for isolation time before the early Solar System.** The ISM ratio for each isotope was computed using equation (2) with $K = 2.3$. The width of the $^{205}$Pb distribution originates in roughly equal parts from the stochastic uncertainty in the ISM ratio and the uncertainty in the meteoritic value (the standard early Solar System (ESS) value from ref. 9 in cyan versus the carbonaceous chondrites (CC) ESS value from ref. 10 in blue). The meteoritic values for $^{107}$Pd/$^{108}$Pd and $^{182}$Hf/$^{180}$Hf are better constrained ($1\sigma \simeq 2$–3%) than that of $^{205}$Pb/$^{204}$Pb (20–33%), so only the stochastic uncertainty is notable. (The $^{135}$Cs/$^{133}$Cs ratio is discussed in Methods only, as $^{135}$Cs has a substantially (approximately 10–20 times) shorter half-life than the three isotopes shown here and needs to be treated differently).

with a much longer isolation time than other star-forming scenarios, has withstood the test from $^{205}$Pb. Improving the $^{205}$Pb/$^{204}$Pb ratio derived for the early Solar System will transform $^{205}$Pb from a short-lived radionuclide that is consistent to one that can authoritatively constrain possible scenarios for the birth of our Sun. Furthermore, the agreement we found between our predicted initial $^{205}$Pb abundance and the range of values inferred from carbonaceous chondrite meteorites[10] supports the use of the $^{205}$Pb–$^{205}$Tl decay system to study the early Solar System chronology of processes that can produce variability in the Pb/Tl ratios. These processes include evaporation owing to thermal processing, crystallization of asteroid cores and spatial separation (differentiation) of different elements inside planets.

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

[1]TRIUMF, Vancouver, British Columbia, Canada. [2]Department of Physics and Astronomy, University of British Columbia, Vancouver, British Columbia, Canada. [3]GSI Helmholtzzentrum für Schwerionenforschung GmbH, Darmstadt, Germany. [4]School of Physics and Astronomy, The University of Edinburgh, Edinburgh, UK. [5]Max-Planck-Institut für Kernphysik, Heidelberg, Germany. [6]Institute of Modern Physics, Chinese Academy of Sciences, Lanzhou, China. [7]Institut für Kernphysik (Theoriezentrum), Fachbereich Physik, Technische Universität Darmstadt, Darmstadt, Germany. [8]Department of Experimental Physics, University of Szeged, Szeged, Hungary. [9]Konkoly Observatory, HUN-REN CSFK, Budapest, Hungary. [10]MTA Centre of Excellence, Budapest, Hungary. [11]E.A. Milne Centre for Astrophysics, University of Hull, Hull, UK. [12]Osservatorio Astronomico di Capodimonte, INAF, Napoli, Italy. [13]I. Physikalisches Institut, Justus-Liebig-Universität Gießen, Gießen, Germany. [14]Osservatorio Astronomico d'Abruzzo, INAF, Teramo, Italy. [15]INFN Sezione di Perugia, Perugia, Italy. [16]II. Physikalisches Institut, Justus-Liebig-Universität Gießen, Gießen, Germany. [17]Department of Physics and Astronomy, University of Victoria, Victoria, British Columbia, Canada. [18]Deutsches Elektronen-Synchrotron DESY, Hamburg, Germany. [19]Physics Department, Technische Universität München, Garching, Germany. [20]Friedrich-Schiller-Universität Jena, Jena, Germany. [21]School of Physics and Astronomy, Monash University, Clayton, Victoria, Australia. [22]ARC Centre of Excellence for All Sky Astrophysics in 3 Dimensions (ASTRO-3D), Melbourne, Australia. [23]Kavli Institute for the Physics and Mathematics of the Universe, University of Tokyo, Kashiwa, Japan. [24]Department of Physics, Panjab University, Chandigarh, India. [25]Nuclear Physics Division, Institute of Research into the Fundamental Laws of the Universe, CEA, Université Paris-Saclay, Gif-sur-Yvette, France. [26]Helmholtz Forschungsakademie Hessen für FAIR (HFHF), GSI Helmholtzzentrum für Schwerionenforschung GmbH, Darmstadt, Germany. [27]Institute of Physics and Astronomy, ELTE Eötvös Loránd University, Budapest, Hungary. [28]Department of Physics and Astronomy, Clemson University, Clemson, SC, USA. [29]J.W. Goethe-Universität, Frankfurt, Germany. [30]Los Alamos National Laboratory, Los Alamos, NM, USA. [31]FH Aachen - University of Applied Sciences, Aachen, Germany. [32]Helmholtz Forschungsakademie Hessen für FAIR (HFHF), GSI Helmholtzzentrum für Schwerionenforschung GmbH, Gießen, Germany. [33]High Energy Nuclear Physics Laboratory, RIKEN, Wakō, Japan. [34]Institut des NanoSciences de Paris, CNRS, Sorbonne Université, Paris, France. [35]Department of Physics, Saitama University, Saitama, Japan. [36]Present address: Institute of Particle and Nuclear Physics, Charles University, Prague, Czech Republic. [37]Present address: Physics Department, Technische Universität München, Garching, Germany. ✉e-mail: guy.leckenby@gmail.com; r.chen@gsi.de; y.litvinov@gsi.de; maria.lugaro@csfk.org

# Methods

## $Q$ value for $^{205}$Tl$^{81+}$

The $Q$ value for the bound-state $\beta^-$ decay of $^{205}$Tl$^{81+}$ is given by

$$Q_{\beta_b}(81 + \rightarrow K, E^*) = -Q_{EC} - E^* - |\Delta B_e| + B_K = 31.1(5) \text{ keV}. \quad (3)$$

The $Q$ value of the electron-capture decay of the ground state of neutral $^{205}$Pb is $Q_{EC} = 50.6(5)$ keV (ref. 26). The energy of the first excited state of $^{205}$Pb is $E^* = 2.329(7)$ keV (ref. 27). The difference in the total atomic binding energy between Tl and Pb is $\Delta B_e = 17.338(1)$ keV and the effective ionization energy of the K shell of bare $^{205}$Pb$^{82+}$ is $B_K = 101.336(1)$ keV (refs. 28,55–57). All uncertainties are $1\sigma$ Gaussian.

## Experimental details

We would like to emphasize that the production and storage (for extended periods of time) of fully ionized $^{205}$Tl beams is only possible at present at the GSI facilities in Darmstadt. Because $^{205}$Tl is stable and abundant on Earth, the easiest solution would be to directly produce a primary beam from an ion source, as was done in the first bound-state $\beta^-$-decay studies on 'stable' $^{163}$Dy (ref. 58) and $^{187}$Re (ref. 59). However, owing to its poisonous vapour, using thallium at GSI is not permitted. Various approaches have been investigated since the 1990s, such as installing a dedicated single-use source, but all were found to be impractical. Hence, the only solution was to produce a secondary beam of $^{205}$Tl in a nuclear reaction. This production process was demonstrated in ref. 60 by creating $^{207}$Tl$^{81+}$ from a $^{208}$Pb beam; however, the investigators required much lower beam intensity than the present experiment and, because of a much higher $Q$ value, were able to observe contaminants directly. Our use of a secondary beam introduces serious complications compared with the methods used in refs. 58,59—whose measurement methods are more directly comparable—owing to the production of daughter contaminants that are mixed with the parent beam.

**Production and separation of $^{205}$Tl$^{81+}$ ions.** According to varying predictions in the literature[35–39], the experiment was planned to be sensitive to the bound-state $\beta^-$-decay half-life of $^{205}$Tl of up to one year. This required at least roughly $10^6$ stored, fully ionized $^{205}$Tl$^{81+}$ ions per measurement cycle. Only recently have the advances in ion source technology and a thorough optimization of the GSI accelerator chain, which includes the linear accelerator UNILAC and the heavy-ion synchrotron SIS-18, enabled accelerated lead beams with a reasonably high intensity of $2 \times 10^9$ particles per spill.

A sample of enriched $^{206}$Pb was used in the ion source. $^{206}$Pb beams were accelerated by the SIS-18 to relativistic energies of 678 MeV per nucleon. This energy was specifically selected to enable stochastic cooling in the ESR (see below). After acceleration, $^{206}$Pb beams were extracted from the SIS-18 within a single revolution, yielding 0.5-μs bunches that were transported to the entrance of the FRS[34]. Here they were impinged on a production target composed of 1,607 mg cm$^{-2}$ of beryllium with 223 mg cm$^{-2}$ of niobium backing. The niobium was used to facilitate the production of fully stripped ions, which dominated the charge-state distribution. All the matter used in the FRS was thick enough to assume that the emerging ions followed equilibrium charge-state distributions[61].

In the projectile fragmentation nuclear reaction, numerous fragments are created by removing nucleons from the projectile. The corresponding cross-sections rapidly decrease with the number of removed nucleons[62]. The primary challenge for our experiment was to eliminate the daughter ions of the studied bound-state $\beta^-$ decay, $^{205}$Pb$^{81+}$, which are amply produced in the reaction through single-neutron removal. All other contaminants were either easily eliminated in the FRS or well separated in the ESR, and were thus not critical.

Owing to the reaction kinematics, as well as energy and angular straggling in the target[63–65], the broad secondary beams of $^{205}$Tl$^{81+}$ and $^{205}$Pb$^{81+}$ ions were indistinguishable after the target. The FRS was tuned such that the beam of $^{205}$Tl$^{81+}$ was centred throughout the separator; see Fig. 2a. At the middle focal plane of the FRS, a wedge-shaped, 735 mg cm$^{-2}$ aluminium energy degrader was placed. The stopping power of relativistic ions in matter depends mostly on their $Z^2$ (ref. 66), and this differential energy loss introduced a spatial separation of $^{205}$Tl$^{81+}$ and $^{205}$Pb$^{81+}$ on the slits in front of the ESR, despite the broad momentum spread of the beams. Using a thicker degrader improved the separation but at the cost of reduced transmission of the ions of interest. Even with this spatial separation, $^{205}$Pb$^{81+}$ ions could not be completely removed and the amount of contamination could only be quantitatively estimated in the offline analysis (see below). Roughly $10^4$ $^{205}$Tl$^{81+}$ ions were injected into the ESR per SIS-18 pulse, with approximately 0.1% $^{205}$Pb$^{81+}$ contamination.

**Cooling, accumulation and storage.** The ions were injected on an outer orbit of the ESR, where the beam was stochastically cooled[67,68]. Outer versus inner orbits of the ESR refers to the wide horizontal acceptance of the ring and can be adjusted by ramping the dipole magnets. Stochastic cooling operates at a fixed beam energy of 400 MeV per nucleon. Hence, the energy of the primary beam was selected such that the $^{205}$Tl$^{81+}$ ions had a mean energy of 400 MeV per nucleon after passing through all the matter in the FRS. A radio-frequency cavity was then used to move the cooled beam to the inner part of the ring, in which several injections were stacked. On the inner orbit, the accumulated beam was continuously cooled by a monoenergetic electron beam produced by the electron cooler[69]. Up to 200 stacks were accumulated. Once the accumulated intensity was sufficient, the beam was moved by the radio-frequency cavity to the middle orbit of the ring, where it was stored for time periods ranging from 0 to 10 h.

The cooling determined the velocity of the ions. Owing to the Lorentz force, the orbit and revolution frequency of the cooled ions were defined only by their mass over charge ($m/q$) ratio. Stored $^{205}$Tl$^{81+}$, $^{205}$Pb$^{81+}$ and $^{205}$Pb$^{82+}$ ions were subject to several processes:
- Recombination in the electron cooler: if a $^{205}$Tl$^{81+}$ or $^{205}$Pb$^{81+}$ ion captured an electron, its charge state was reduced to $q = 80+$ and its orbit was substantially altered, causing it to be lost from the ESR acceptance. Similar electron recombination for $^{205}$Pb$^{82+}$ ions reduced their charge state to $q = 81+$, where they returned to the main beam and remained in the ESR. To minimize the recombination rate, the density of electrons in the cooler during the storage time was set to 20 mA, which was found to be the minimum value to maintain the beam.
- Collisions with the rest-gas atoms: in such collisions, $^{205}$Tl$^{81+}$ and $^{205}$Pb$^{81+}$ ions underwent charge-exchange reactions. If a $^{205}$Tl$^{81+}$ or $^{205}$Pb$^{81+}$ ion captured an electron, it was lost from the ring (as above). If a $^{205}$Pb$^{81+}$ ion lost an electron, it remained stored in the ESR on an inner orbit. Capture of an electron by $^{205}$Pb$^{82+}$ moved it to the main beam at $q = 81+$. Thanks to the ultrahigh vacuum of the ESR, the collision rate was low, as demonstrated by the achieved storage times of up to 10 h.
- Bound-state $\beta^-$ decay of $^{205}$Tl$^{81+}$: this is the process of interest. As noted previously, the mass difference ($Q$ value), between $^{205}$Tl$^{81+}$ and $^{205}$Pb$^{81+}$ is only 31 keV, which meant that both beams completely overlapped in the ESR and remained stored on the central orbit.

The $^{205}$Tl$^{81+}$ loss rate during storage as a result of all of the above processes was determined to be $\lambda_{Tl}^{loss} = 4.34(6) \times 10^{-5}$ s$^{-1}$, corresponding to a beam half-life of 4.4 h. The $^{205}$Pb$^{81+}$ loss rate was determined by a theoretical scaling of the relative radiative recombination rates, resulting in a differential loss rate of $\lambda_{Tl}^{loss} - \lambda_{Pb}^{loss} = 3.47(5)_{stat}(87)_{syst} \times 10^{-6}$ s$^{-1}$.

**Detection.** The $^{205}$Pb$^{81+}$ ions detected at the end of the storage period consisted of both ions created by bound-state $\beta^-$ decay and the contamination transmitted from the FRS. The only way to separate the few $^{205}$Pb$^{81+}$ ions from the vast amount of $^{205}$Tl$^{81+}$ ions was to remove the bound electron from $^{205}$Pb$^{81+}$. This was done by using the supersonic

Ar gas-jet target that is installed in the ESR[70,71]. The density of Ar gas was about $10^{12}$ atoms $cm^{-2}$ and it was switched on for 10 min. During this time, to keep the beam together, the density of the electrons in the cooler had to be increased to 200 mA. Charge-exchange reactions and different recombination rates had to be taken into account; see the analysis details below. $^{205}Pb^{82+}$ ions produced during this stripping were moved to the inner orbit of the ESR, where they were cooled and counted non-destructively.

Several detectors were implemented throughout the experiment:

- A current comparator is an inductive device to measure the total current produced by the stored beam. It is permanently installed at the ESR for diagnostics purposes and is sensitive to beam intensities in excess of about $10^4$ particles. This detector was used to continuously monitor the high-intensity $^{205}Tl^{81+}$ beam assuming that the contribution from all other contaminants was negligible.
- Multiwire proportional chambers are position-sensitive, gas-filled detectors installed in special pockets separated from the ESR vacuum by 25-µm, stainless-steel windows[72]. These detectors were used to count produced $q = 80+$ ions to determine the charge-changing cross-section ratio (see below) and for a complementary measurement of the beam lifetime during the crucial gas-stripping phase.
- A non-destructive Schottky detector was used to continuously monitor frequency-resolved intensities of individual ion species throughout the entire experiment without interruptions. This detector features a cavity of air separated from the ring vacuum by a ceramic gap[73]. Relativistic ions that pass by induce an electric field in the cavity. The ions revolved at about 2.0 MHz, whereas the cavity was resonant at about 245 MHz, meaning that the detector was sensitive to roughly the 125th harmonic. The Fourier transform of the amplified noise from the cavity yielded a noise-power-density spectrum, of which an example spectrum is shown in Extended Data Fig. 1, in which the frequencies of the peaks corresponded to the $m/q$ ratios of the stored ion species[74], whereas the intensities were proportional to the corresponding number of stored ions[75,76]. The Schottky detector had a wide dynamic range, meaning that the detector itself is sensitive to very low as well as very high excitation amplitudes without any distortion, even in the same spectrum. This allows the Schottky detector to monitor millions of ions while still being sensitive to single ions[77,78]. Unfortunately, in this experiment, the detector was saturated for high-intensity beams and had to be cross-calibrated with the current comparator.

## Determination of the bound-state β⁻-decay rate

The number of $^{205}Tl^{81+}$ ions in the ESR decreased exponentially throughout the storage period owing to radiative electron recombination in the electron cooler and charge-changing collisions with the rest-gas atoms, resulting in $^{205}Tl^{80+}$ ions that left the acceptance of the storage ring. The growth of $^{205}Pb^{81+}$ daughters must then be solved with a differential equation: the details are provided in ref. 79. The full solution to the differential equations is

$$\frac{N_{Pb}(t_s)}{N_{Tl}(t_s)} = \left( \frac{N_{Pb}(0)}{N_{Tl}(0)} + \frac{\lambda_{\beta_b}/\gamma}{\lambda_{\beta_b}/\gamma + \lambda_{Tl}^{loss} - \lambda_{Pb}^{loss}} \right)$$
$$\exp((\lambda_{\beta_b}/\gamma + \lambda_{Tl}^{loss} - \lambda_{Pb}^{loss})t_s) \qquad (4)$$
$$- \frac{\lambda_{\beta_b}/\gamma}{\lambda_{\beta_b}/\gamma + \lambda_{Tl}^{loss} - \lambda_{Pb}^{loss}},$$

with the same notation as in equation (1). The storage ring loss rate $\lambda_{Tl}^{loss}$ was determined from the exponential decrease (Fig. 2b shows an example measurement) whereas $\lambda_{Pb}^{loss}$ was scaled using a theoretical calculation from $\lambda_{Tl}^{loss}$. Using a Taylor series expansion and noting that $\lambda_{\beta_b} \ll (\lambda_{Tl}^{loss} - \lambda_{Pb}^{loss})$, this full solution can be well approximated by equation (1), with <0.2% difference over our storage lengths.

The ion intensity inside the storage ring was monitored by the Schottky detector described above, in which the noise-power density integrated over a peak (SA) in the spectrum is directly proportional to the ion number of that species. Thus, the ratio of the number of $^{205}Pb^{81+}$ daughter ions to the number of $^{205}Tl^{81+}$ parent ions is equivalent to the ratio of respective Schottky integrals, after several corrections have been applied:

$$\frac{N_{Pb}(t_s)}{N_{Tl}(t_s)} = \frac{SA_{Pb}(t_s)}{SA_{Tl}(t_s)} \frac{1}{SC(t_s)} \frac{1}{RC} \frac{\varepsilon_{Tl}(t_s)}{\varepsilon_{Pb}(t_s)} \frac{\sigma_{str} + \sigma_{rec}}{\sigma_{str}}. \qquad (5)$$

There are four corrections that need to be implemented:

1. The saturation correction SC corrects for an observed saturation of the Schottky DAQ system[78] at large noise-power densities owing to a mismatched amplifier switch. This correction was determined individually for each measurement by calibrating the observed non-exponential decay against the exponential decay constant measured in the multiwire proportional chamber. The uncertainty in this correction was dominated by the calibration fit, so is a systematic uncertainty.
2. The resonance correction RC accounts for the resonance response of the Schottky detector, which resulted in an amplification of the noise-power density at the $^{205}Tl^{81+}$ frequency when compared with the $^{205}Pb^{82+}$ frequency. This correction was extracted by observing the Schottky area change at the orbit shift after accumulation. Because it is a property of the Schottky detector, the correction was applied globally and is also a systematic uncertainty.
3. The interaction efficiency $\epsilon$ corrects for the number of ions that interacted with the gas target before the Schottky measurement, accounting for the loss of $^{205}Tl^{81+}$ owing to electron recombination and the proportion of $^{205}Pb^{81+}$ that were stripped to the 82+ charge state. This correction was determined from the multiwire proportional chamber event rate and was highly correlated with the gas target density, and so was applied individually. As a result, it contributed to the statistical uncertainty of the measurement.
4. The charge charge-changing cross-section ratio $(\sigma_s + \sigma_r)/\sigma_s$, which corrects for any $^{205}Pb$ daughter ions lost to electron recombination rather than stripping in the gas target. This correction was determined by counting both atomic reaction channels using a $^{206}Pb^{81+}$ beam. This is a physical constant and so was applied globally, contributing to the systematic error.

Full details on these corrections are discussed in refs. 79,80 and in the upcoming thesis of G. Leckenby. Intermediate and result data after these corrections have been applied are available in ref. 81.

## Estimated contamination variation

One source of error that could not be independently determined was the variation in the amount of contaminant $^{205}Pb^{81+}$ ions injected into the storage ring from the projectile fragmentation reaction. The presence of the contamination is obvious from the non-zero $t = 0$ intercept, as seen in Extended Data Fig. 2, but variation in that contamination is impossible to measure and account for without purging the $^{205}Tl^{81+}$ beam using the gas target, which would reduce intensities and hence the accumulated signal. Initially, we expected any variation in the contaminant yield to be negligible. However, by cutting away everything but the extreme tails of the $^{205}Pb^{81+}$ fragmentation distribution, the impact of instabilities in the yield becomes notable. The presence of unaccounted uncertainty in the data is obvious, both visually when considering the residuals in Extended Data Fig. 2a and noting that the 95% confidence interval for 14 degrees of freedom is $\chi^2 = [6.6, 23.7]$, whereas our data have $\chi^2 = 303$. We have exhausted all other possibilities of stochastic error and thus conclude that we must estimate the variation of contaminant $^{205}Pb^{81+}$ from the data itself.

Appealing to the central limit theorem, we assume that the contamination variation is normally distributed. To estimate the missing uncertainty from the data, the $\chi^2(\nu = 14)$ distribution was sampled for each Monte Carlo run and then a value for the missing uncertainty was determined by solving the following $\chi^2$ for our data:

$$\chi^2 = \sum_i \frac{(\text{data}_i - \text{model}_i)^2}{\sigma_{i,\text{stat}}^2 + (\exp[(\lambda_{\text{Tl}}^{\text{loss}} - \lambda_{\text{Pb}}^{\text{loss}})t_s] \times \sigma_{\text{CV}})^2}, \tag{6}$$

in which $\sigma_{\text{CV}}$ is the estimated contamination variation and $\sigma_{\text{stat}}$ is the statistical uncertainty from all other sources. Note that the growth factor $\exp[(\lambda_{\text{Tl}}^{\text{loss}} - \lambda_{\text{Pb}}^{\text{loss}})t_s]$ is included to account for how the initial contamination evolves with storage time. This growth factor is required to ensure that the terms of the sum follow a unit normal distribution to satisfy the requirements of a $\chi^2$ distribution. Thus, for each iteration of the Monte Carlo error propagation, a different value of $\chi^2$ is used to estimate the missing uncertainty to account for the stochastic nature of the distribution. The code for this Monte Carlo error propagation is available in ref. 82.

The error-propagation method described above was double-checked by performing a Bayesian analysis considering the systematic uncertainties as prior distributions[83–85], which confirmed our Monte Carlo method within the quoted uncertainties.

### $^{205}$Pb and $^{205}$Tl weak rates calculation

The bound-state β$^-$-decay rate, $\lambda_{\beta_b}$, of fully ionized $^{205}$Tl$^{81+}$ with the production of an electron in the K shell is given by

$$\lambda_{\beta_b} = \frac{\ln(2)}{\mathcal{K}} C_K f_K, \tag{7}$$

with $\mathcal{K} = 2\bar{F}t = 6144.5(37)$ s the decay constant determined by measurements of super-allowed β decay[86], $f_K = \pi Q_{\beta_b}^2 \beta_K^2 \mathcal{B}_K / 2m_e^2$ the phase space for bound β$^-$ decay with $Q_{\beta_b}$ the $Q$ value given in equation (3), $m_e$ the electron mass, $\beta_K$ the Coulomb amplitude of the K-shell electron wavefunction and $\mathcal{B}_K$ the exchange and overlap correction[87]. Using $\beta_K^2 \mathcal{B}_K = 5.567$ for hydrogen-like $^{205}$Pb$^{81+}$ computed with the atomic code from ref. 88, we have $f_K = 0.032(1)$, which—together with the measured decay rate—gives a value for the nuclear shape factor for bound β$^-$ decay $C_K = 7.6(8) \times 10^{-3}$, corresponding to $\log(ft) = \log(\mathcal{K}/C_K) = 5.91(5)$.

Following the β-decay formalism of refs. 87,89, the nuclear shape factor can be expressed as a combination of different first-forbidden matrix elements. Although the value of the matrix elements connecting the $^{205}$Tl(1/2$^+$) and $^{205}$Pb(1/2$^-$) states is independent of the weak process considered, they appear in different combinations for bound β$^-$ decay of $^{205}$Tl and continuous and bound-electron capture of $^{205}$Pb. To disentangle the individual nuclear matrix elements, we have performed shell-model calculations using the code NATHAN[90] and the Kuo–Herling interaction[40] (for details, see R.M., T.N. & G.M.-P., manuscript in preparation).

Depending on the stellar conditions, $^{205}$Tl and $^{205}$Pb ions will be present in different ionization states. To determine their population, we follow the procedure in ref. 43. However, we have revised the treatment of the Coulomb energy of the ion in the stellar plasma. We treat the multicomponent stellar plasma within the additive approximation, that is, all of the thermodynamic quantities are computed as a sum of individual contributions for each species. Furthermore, we assume that the electron distribution is not affected by the presence of charged ions (uniform background approximation). Under these approximations, the energy of the ion in the stellar plasma can be obtained by[91]

$$\mathcal{E}(Z_i) = \mathcal{E}_0(Z_i) + \mu_C, \quad \mu_C = k_B T f_C(\Gamma_i), \quad \Gamma_i = \frac{Z_i^{5/3} e^2}{a_e k_B T}, \quad a_e = \left(\frac{3}{4\pi n_e}\right)^{1/3} \tag{8}$$

with $\mathcal{E}_0$ the energy of the ion in vacuum, $Z_i$ the net charge of the ion, $n_e$ the electron density and $f_C(\Gamma_i)$ the Coulomb free energy per ion in units

of $k_B T$ that we approximate following equation (2.87) in ref. 92. We note that, in our approximation, the Coulomb energy of an ion in the stellar plasma depends only on the net charge of the ion and is independent of the internal structure of the ion. Hence, all states with the same net charge are corrected in the same way. Under this approximation, Coulomb corrections only affect processes in which the net charge of the ion is modified. This includes ionization and continuous electron capture, whereas bound-electron capture and bound β$^-$ decay are not modified. We differ in the treatment of the latter from ref. 43. The effective ionization energy of a specific ionic state in the stellar plasma is reduced by an amount $\Delta\chi(Z_i) = \mu_C(Z_i + 1) - \mu_C(Z_i)$ (we notice that $\mu_C$ is negative with our definition and grows in magnitude with increasing $Z_i$). This reduction is denoted as depletion of the continuum in ref. 43. Similarly, the $Q$ value for continuous electron capture on an ion with net charge $Z_i$ changes by an amount $\Delta Q_C = \mu_C(Z_i) - \mu_C(Z_i - 1)$. After accounting for these corrections, the different stellar weak processes are computed using the standard expressions (see, for example, ref. 43).

Extended Data Fig. 3 compares the weak rates connecting $^{205}$Pb and $^{205}$Tl for two different electron densities, $n_e = 10^{25}$ cm$^{-3}$ and $n_e = 10^{27}$ cm$^{-3}$, as a function of temperature. We find that electron-capture processes on $^{205}$Pb are dominated by bound-electron capture except at very high densities $n_e \gg 10^{27}$ cm$^{-3}$. At very low temperatures, the capture rate approaches the laboratory value $\lambda_{ec} = 4.1(2) \times 10^{-8}$ year$^{-1}$ plus a correction owing to continuous electron capture at high electron densities. With increasing temperature, the rate increases as a result of the thermal population of the 1/2$^-$ excited state of $^{205}$Pb. Bound-electron capture proceeds mainly from L-shell electrons and it is suppressed once the temperature is high enough for $^{205}$Pb to be at ionization states for which the L-shell orbits are empty, $T \gtrsim 50$ MK. At these conditions, holes in the K shell start to appear and bound β$^-$ decay of $^{205}$Tl becomes the dominating weak process once the temperature reaches $T \gtrsim 100$ MK.

### Revised $(n, \gamma)$ cross-sections

Recommended $(n, \gamma)$ cross-sections for s-process energies ($kT = 5$–100 keV) are available as Maxwellian-averaged cross-sections for nuclei in the ground-state from the KADoNiS database[93]. The available version 0.3 (ref. 94) was last updated around 2009. A partial, however incomplete, update to KADoNiS v1.0 was done in 2014 (ref. 48). For this publication, the neutron-capture cross-sections of nine isotopes were revisited and new recommended values with the latest experimental data were provided (Extended Data Table 1). This included the stable isotopes $^{202}$Hg, $^{204}$Hg, $^{203}$Tl, $^{205}$Tl, $^{204}$Pb and $^{206}$Pb, as well as the radioactive isotopes $^{203}$Hg, $^{204}$Tl and $^{205}$Pb. For the stellar-abundance calculations, the recommended Maxwellian-averaged cross-section values have to be multiplied by the (temperature-dependent) stellar enhancement factor (SEF) to simulate the impact of the population of excited states in a stellar plasma. These values are listed for each isotope in the KADoNiS v1.0 database but, for ease of access, we give the SEF of the nine isotopes here discussed in Extended Data Table 1.

It should be emphasized that, to identify whether a given cross-section measured in the laboratory (in the ground state) can also help constrain the stellar cross-section (captures from excited states), Rauscher et al.[95] have introduced the ground-state contribution $X$. This factor $X$ is also given in the latest KADoNiS version and is shown in Extended Data Table 1. A large deviation from 1 implies that the (unmeasured) contributions from excited states have a larger impact on the stellar cross-section.

For the six stable nuclei, revised experimental information was included as follows:

- $^{202}$Hg: the $kT = 30$ keV activation data and its uncertainty[96] has been renormalized by $f = 1.0785$ to the new $^{197}$Au$(n, \gamma)^{198}$Au value at this energy and extrapolated with the energy dependencies from the JEFF-3.1 (ref. 97), JENDL-3.3 (ref. 98) and ENDFB/VII.1 (ref. 99) libraries.

- $^{204}$Hg: same procedure as for $^{202}$Hg (ref. 96) but the experimental uncertainty of 47% was used for the whole energy range. The libraries JEFF-3.3 (ref. 100) and ENDF/B-VIII.0 (ref. 101) were excluded, as they show unphysical trends at energies below 1 keV. Only the energy dependencies of TENDL-2019 (ref. 102) and JEFF-3.0A (ref. 103) were used for the extrapolations.

- $^{203}$Tl: the new recommended values are an average of recently evaluated data libraries (TENDL-2019 (ref. 102), JEFF-3.3 (ref. 100), JEFF-3.0A (ref. 103) and ENDF/B-VIII.0 (ref. 101)). These libraries include the only available experimental time-of-flight data from 1976. The uncertainty is estimated as the standard deviation between the four libraries.

- $^{205}$Tl: only the ENDF/B-VIII.0 (ref. 101) data reproduce previous measurements and were used for the recommendation. A 25% uncertainty was assumed for the whole energy region.

- $^{204}$Pb: the new recommended values are based on the time-of-flight measurement by ref. 104 and have been included in JENDL-4.0 (ref. 105) over the whole energy range. An uncertainty of 5% was assumed, slightly higher than the uncertainties of 3.0–4.4% from the experiment.

- $^{206}$Pb: the new recommended values are based on the two time-of-flight measurements[106,107] up to $kT = 50$ keV and the respective uncertainty was used. Beyond that energy, an average of recently evaluated data libraries (JEFF-3.3, JENDL-4.0, JEFF-3.0A and ENDF/B-VIII.0) gives a good representation, and an uncertainty of 7% was used for $kT = 50$–100 keV.

For the three radioactive $N = 123$ isotones $^{203}$Hg ($t_{1/2} = 46.594$ days), $^{204}$Tl ($t_{1/2} = 3.783$ years) and $^{205}$Pb ($t_{1/2} = 17.0$ Myr), the KADoNiS database could, so far, only provide 'semiempirical' estimates because no experimental data existed. The n_TOF collaboration has now measured $^{204}$Tl$(n, \gamma)$ for the first time[108]. The new experimental data are a factor of 2 lower than the values given by TENDL-2019, ENDF/B-VIII.0 and JEFF-3.3, and a factor of up to 2 higher than the TENDL-2021 and JEFF-3.0A values. This shows the importance of replacing theoretical values with experimental data when available, especially for astrophysical model calculations.

For $^{203}$Hg and $^{205}$Pb, for which no experimental information exists, the best approach is to take the average of the most recently revised (recalculated) cross-section libraries and assign a large uncertainty, commonly the standard deviation between the libraries. The $(n, \gamma)$ cross-sections for the isotopes of interest for each of these libraries have been investigated, and those with unexplained 'nonphysical' trends (such as, for example, for JEFF-3.3 and ENDF/B-VIII.0 in the case of $^{204}$Hg) have been excluded for the calculation of the averaged cross-section. For the recommended $^{203}$Hg and $^{205}$Pb cross-sections, the libraries used were ENDF/B-VIII.0, JEFF-3.3, TENDL-2019 and TENDL-2021. However, given the large deviations between the libraries, these values should be better constrained as soon as possible with experimental data.

The new recommended Maxwellian-averaged cross-section for $kT = 5$–100 keV for the nine discussed isotopes are given in Extended Data Table 1. The listed SEFs and $X$ factors have been extracted from the KADoNiS database[48] and are also given for completeness, but these values have not been changed.

## The $^{205}$Pb/$^{204}$Pb ratio in the early Solar System

The method to extract isotopic ratios of short-lived radioactive isotopes relative to a stable, or long-lived, isotope of the same element at the time of the formation of the first solids in the early Solar System is founded on chemistry. It is based on a linear regression between, on the $y$ axis, the measured ratio of the daughter nucleus relative to another stable isotope of the same element (for example, $^{205}$Tl/$^{203}$Tl) and, on the $x$ axis, the ratio of a stable isotope of the same element as the short-lived radioactive isotope relative to the same denominator as the $y$ axis (for example, $^{204}$Pb/$^{203}$Tl). Data points from the same meteorite, or meteoritic inclusion, will sample material with a variety of $^{204}$Pb/$^{203}$Tl ratios, depending on their chemistry. If $^{205}$Tl/$^{203}$Tl varies with $^{204}$Pb/$^{203}$Tl, then it can be concluded that the correlation is driven by the decay of $^{205}$Pb, as this isotope will chemically correlate with $^{204}$Pb. The slope of this correlation line (also referred to as isochrone, as all the data points lying on it would have formed at the same time) provides the $^{205}$Pb/$^{204}$Pb ratio at the time of the formation of the sample material (meteorite or inclusion). The initial value in the early Solar System can be derived by reversing the radioactive decay of the ratio using the age difference between the sample material and the first solids, that is, the oldest meteoritic calcium–aluminium inclusions. The sample ages can be derived using other radiogenic systems, such as U–Pb.

Although the method is robust, the variations to be measured are so small (in the case of $^{205}$Tl/$^{203}$Tl, they may be on the third or fourth significant digit) that the handling of the uncertainties and the removal of isotopic variations owing to effects other than the radiogenic contribution becomes particularly crucial. Among such variations, the most prominent are those resulting from the chemical effects that depend on the mass of the isotope. These can usually be removed by internal calibration; however, this requires at least three isotopes to be measured. This is not possible for either Tl, as it only has two stable isotopes, or Pb, because three out of its four stable isotopes are affected by radiogenic contributions from U–Th decay chains. Furthermore, the original Pb abundance in the sample is easily contaminated by anthropogenic Pb. Because of these difficulties, it was not possible to derive robust $^{205}$Pb/$^{204}$Pb ratios in the early Solar System until the 2000s. Since then, three studies have attempted to obtain reliable data from iron meteorites[11,109] and carbonaceous chondrites[10]. Reference 10 also measured the Pb and Cd isotopic compositions of the meteorites and ref. 11 also measured Pt. Because Cd and Pt behave similarly to Tl from the point of view of chemistry, these data allowed the identification and therefore elimination of samples affected by mass-fractionation processing. Furthermore, ref. 10 also measured the Pb isotopic compositions to correct for terrestrial Pb contamination.

The carbonaceous chondrites data[10] resulted in an isochrone with slope $(1.0 \pm 0.4) \times 10^{-3}$ (at $2\sigma$). This is taken to be representative of the early Solar System because these meteorites are believed to record nebular processes. The analysed iron meteorites instead record later formation times, typically 10–20 Myr later (which means that the slope of their isochrone is, by definition, lower than that of the carbonaceous chondrites), and—by evaluating different age determinations—it is possible to establish whether the different data are consistent with each other. The value measured by the isochrone of ref. 109 requires much longer formation times (on the order of 60 Myr) or, alternatively, a much lower initial value, by roughly a factor of 10, than that derived by ref. 10. The value measured by ref. 11 instead provides more consistent ages, in agreement with the initial value of ref. 10. However, the $y$-axis intercept of the isochrone of ref. 11, that is, at the zero value of $^{204}$Pb/$^{203}$Tl, is lower by a few parts per ten thousand than that of ref. 10. This prompted the suggestion that the actual slope of the carbonaceous chondrites data should be higher, that is, $(2 \pm 1) \times 10^{-3}$ (at $2\sigma$), such that its intercept would the same as the new iron meteorite data. Given these inherent uncertainties, it was suggested by ref. 9 to use an initial value that covers the range of the two studies, that is, $(1.8 \pm 1.2) \times 10^{-3}$ (at $2\sigma$). We have used both the range suggested by ref. 9 and the original unmodified slope from carbonaceous chondrites reported in ref. 10.

The previous predicted AGB upper limit for the $^{205}$Pb/$^{204}$Pb ISM ratio of $5 \times 10^{-4}$ (ref. 15) is in contradiction with (that is, it is lower than) the most recent laboratory data. Our new predicted ISM value resolves this tension, as it is roughly an order of magnitude higher, although the two values are not directly comparable with each other. In fact, the previous upper limit represents the ratio expected from the ejecta of one single AGB star only and without the inclusion of the main ($^{13}$C$(\alpha, n)^{16}$O) neutron source, therefore, of an AGB star that would not produce $s$-process isotopes. The original aim was to avoid overproduction of all the $s$-process short-lived isotopes (especially $^{107}$Pd) relative to $^{26}$Al in the

scenario in which a single AGB star located near the early Solar System would have contributed all these radioactive isotopes (see also ref. 110). Our results and those from ref. 18 show that, instead, the *s*-process isotopes have a separate origin from [26]Al: they are all self-consistently explained by the chemical evolution of the Galaxy driven by the material ejected by many different AGB stars, in agreement with the latest [205]Pb/[204]Pb laboratory meteoritic analysis. Furthermore, because our results generally agree better with the lowest values of the range recommended at present, they support the value derived from the slope of the original carbonaceous chondrites isochrone.

## Yields from AGB star models

The AGB models were calculated to simulate *s*-process nucleosynthesis in these stars (as described in detail in ref. 3) using a revised version of the Monash nucleosynthesis tools[49,111], which allow detailed incorporation of the temperature and density of β-decay and electron-capture rates. The Monash nucleosynthesis code is a post-processing tool, which acts on a nuclear network coupled to stellar structure inputs generated by the Monash stellar evolution code. The post-processing method is relatively fast and works under the assumption, valid here, that the reaction rates under investigation do not contribute to the bulk of the stellar energy generation. The nucleosynthesis code simultaneously solves the changes owing to nuclear burning and to convection, implemented through an advective scheme. Specifically, this means that, within convective regions (that is, the thermal pulses and the envelope of the star), [205]Tl and [205]Pb decay, while at the same time they are mixed through different stellar layers of different temperature and densities. The relevant $(n, \gamma)$ rates were included as described above and, when compared with models using previous values of these rates, the differences were on the order of 10% or less. The rate of the debated neutron source $^{22}$Ne$(\alpha, n)^{25}$Mg was taken from ref. 112; see also discussion in ref. 113. Using the lower rate in ref. 114 resulted in less than 10% difference.

To determine the yield of a population of AGB stars at the time of the formation of the Sun, we considered the ejecta from stars of masses 2.0–4.5 $M_\odot$, that is, those expected to contribute towards *s*-process element production in the Galaxy[115], for an initial composition that is the same as the proto-solar nebula, in which $Z_\odot = 0.014$ (ref. 116). We also tested the case in which the initial metallicity of the AGB stars is $Z = 0.02$, as discussed further below. The resulting yields, that is, the total ejected mass of the indicated isotope and their ratios, are listed in Extended Data Table 2. The [205]Pb/[204]Pb ratio shows the main effect of temperature on the production of [205]Pb. Increasing the stellar mass, the temperature also increases: the maximum temperature reached in the thermal pulse increases from 280 to 356 MK for the mass range considered in Extended Data Table 2. This means that, in the higher-mass stars, during the activation of the $^{22}$Ne neutron source, [205]Tl and [205]Pb experience stronger and weaker decays, respectively (see Fig. 3, noting that the most relevant electron density for the intershell of AGB stars is around the $10^{27}$ cm$^{-3}$ value, that is, on the order of 3,000 g cm$^{-3}$). As described in the main text, the two isotopes will continue to decay after the thermal pulse is extinguished and before they are dredged up to the envelope. The exact effect of this phase depends on the detailed temperature and density structure of the region, as well as the time that elapses between the thermal pulse and the following dredge-up. The average mass yield ratio of this AGB stellar population is 0.168 (0.167 by number abundance) when using the trapezoidal rule to integrate the yields over Salpeter's initial mass function. In our models, stars less than 2 $M_\odot$, at this metallicity, do not eject *s*-process elements[111]; however, this result is model-dependent. We tested the most conservative scenario of extending the range of masses down to 1.5 $M_\odot$ by assuming the same [204]Pb yield as the 2 $M_\odot$ model and no ejection of [205]Pb, owing to the colder temperature. Even in this extreme case, the average yield ratio decreases by only less than 10%. Similarly, if we extended our mass grid to reach masses of 6 $M_\odot$, in the conservative case in which they

ejected the same amounts of $^{204,205}$Pb as the 4.5 $M_\odot$ model, we would obtain an increase of the final ratio by 10%. Overall, AGB stars with masses beyond the range considered here would not have a substantial impact on our results.

Differences appear when comparing AGB models calculated using different evolutionary codes. This is mostly because of the fact that different codes produce stellar models with different temperatures, which—as seen above—has the greatest impact on the final results. To perform this analysis quantitatively, we computed a 3 $M_\odot$ model of metallicity $Z = 0.02$ using the Monash, FUNS and NuGrid tools. The FUNS models have been calculated with the most recent version of the code, which includes mixing induced by magnetic fields[50,117]. These models use as a reference the solar mixture published by Lodders[118], with updates from ref. 119. In the FUNS models, the nucleosynthesis is directly calculated with the physical evolution of the structure, thus no post-processing technique is applied. The NuGrid models are based on the stellar structure computed[51] with the stellar evolution code MESA[120] including a convective boundary mixing prescription at the border of convective regions[121]. The solar distribution used as a reference is given in ref. 122. The detailed nucleosynthesis is calculated using the stellar structure evolution data as input for a separate post-processing code[123]. The FUNS results provided a [205]Pb/[204]Pb ratio of 0.021, roughly a factor of 3 lower than the corresponding Monash ratio of 0.071. In the case of NuGrid, instead, the adopted convective boundary mixing prescription results in higher temperatures and, in turn, a higher [205]Pb/[204]Pb ratio of 0.176. With the Monash code, we also tested implementing different opacities and initial abundances (to mimic the choices made in the other codes) and the results were affected by less than 10%. Therefore, the overall variation of roughly a factor of 10 between the three different models is most probably because of: (1) the inclusion of overshoot at the base of the thermal pulse in the NuGrid models, which results in higher temperatures than the other models, and (2) the different mass-loss rates implemented: ref. 124 in Monash, ref. 125 in NuGrid and ref. 126 in the FUNS model.

## Radioactive nuclei in GCE

The calculation of the ISM abundance ratio [205]Pb/[204]Pb according to equation (2) includes a factor, $K$, which allows us to account for the impact of various galactic processes. As described in detail previously[53], current observations can be used to constrain models of the Milky Way galaxy, including the gas inflow rate, the mass of gas, the star formation rate, the mass of stars and the core-collapse supernova and Type Ia supernova rates. It is therefore possible to produce several realizations of the Milky Way galaxy that reproduce the observed ranges of such properties, and each of these realizations will result in a different radioactive-to-stable isotope ratio. After analysis of the possible effects, ref. 53 provided a lower limit, a best fit and an upper limit for the value of $K$ of 1.6, 2.3 and 5.7, respectively, which can be used in equation (2) to account for galactic uncertainties. In the main text, we have focused on the best fit $K = 2.3$ case; here in Extended Data Fig. 4 and Extended Data Table 3, we also show the results using the upper and lower limits. Note that each value of $K$ represents a different realization of the Milky Way galaxy, therefore time intervals can only be compared with each other when they are calculated using the same $K$.

The use of equation (2) is not as accurate as a full GCE model because it allows for only one stellar production ratio, whereas this number varies with stellar mass and metallicity. To check its validity, we tested the results of using equation (2) for [107]Pd/[108]Pd, [135]Cs/[133]Cs and [182]Hf/[180]Hf against those of the GCE models[18]. We found that the steady-state equation reproduces the more accurate, full GCE simulations that include variable yields within 50%. Furthermore, the production ratios $P$ calculated from AGB stars are *s*-process production ratios. As noted in the main text, the contribution of live *r*-process abundances to the *s*-process short-lived radionuclides is negligible. However, the *s*-process production ratio $P$ must be scaled to account for the *r*-process contribution

to the stable reference isotope. We use the s-process fraction of the stable reference calculated for the Monash GCE models provided in ref. 18. To do this, we multiply the s-process production ratios by the s-process fraction of the stable reference calculated for the Monash GCE models provided in ref. 18.

All of the distributions plotted in Extended Data Fig. 4 also include the uncertainties in the steady-state value owing to the fact that stellar ejections are not continuous but discrete events, with a time interval competing with the decay time. We calculated these uncertainties by running simulations with the Monte Carlo code developed in ref. 54, in which a stellar ejection event consists of injecting a unit of material into a parcel of interstellar gas with the intent to simulate the enrichment of that parcel with radioactive isotopes from one or many AGB star sources. According to the full analysis of ref. 54, the steady-state assumption is valid for this process if the ratio of the mean life $\tau$ and the interval $\delta$ that elapses between each injection event is greater than 2. Therefore, for $^{107}Pd/^{108}Pd$ and $^{182}Hf/^{180}Hf$, we used the same choice of parameters as ref. 18, that is, the most conservative choice $\delta \simeq 3$ Myr and $\tau/\delta \approx 3$–4. Given its longer mean life, this assumption is also satisfied for $^{205}Pb$. Physically, AGB winds may not have enough energy to be able to carry material far enough from the source to realize the relatively short $\delta$ assumed here (a simple calculation of $\delta$ based on energy conservation would instead give values on the order of 50 Myr (ref. 1)). However, other processes, such as core-collapse supernova shock waves[127] and diffusion[128,129], probably contribute to further spreading of AGB material in the Galaxy, thereby allowing it to reach more parcels of gas in shorter time intervals.

The shorter mean life of $^{135}Cs$ means that this isotope would be in steady-state equilibrium only if $\delta \simeq 1$ Myr, in which case we can derive lower limits for the corresponding isolation time, which are shown in Extended Data Fig. 4. (Note that, for $\delta \simeq 3$ Myr, only an upper limit of the $^{135}Cs$ abundance can be derived; see Table 4 of ref. 54. As an upper limit is also only available for the early Solar System, the isolation time is undefined in this case). The new values for the isolation time are shorter than those provided in ref. 18. This is because of the combined effect of the revised $\tau$ used here (1.92 Myr), which is 70% lower than the value used in ref. 18 (3.3 Myr), and the roughly two times higher production ratio of $^{135}Cs/^{133}Cs$, owing to the new rate of the decay of $^{134}Cs$ (refs. 130,131), the branching point leading to the production of $^{135}Cs$.

When the value of $K$ increases, all of the radioactive-to-stable isotope ratios increase, according to equation (2). Therefore, as shown in Extended Data Fig. 4, the isolation time also increases and the increase is proportional to the mean life of each isotope, which is why the shift is the largest for the $^{205}Pb$ distribution. The overlap between the three distributions is the largest for $K = 5.7$. If we assume that the $^{205}Pb/^{204}Pb$ average mass yield ratio varies according to the results of the FUNS and NuGrid models discussed in the previous section (that is, /3.4 and ×2.5, respectively, relative to the Monash models), then the $^{205}Pb/^{204}Pb$ time distributions in Extended Data Fig. 4 shift by −30 Myr and +22 Myr, respectively. These variations call for a more detailed future analysis of the production of the four s-process short-lived isotopes in different AGB models. The s-process $^{107}Pd/^{108}Pd$ production ratio is typically ≃0.14, as it is controlled by the ratio of the neutron-capture cross-sections of the two isotopes, which are relatively well known[132,133]. Therefore, the main challenge for nuclear-physics inputs remain for the $^{182}Hf/^{180}Hf$ ratio, which is controlled by activation of the temperature-dependent branching point at $^{181}Hf$, a function of the decay rate of $^{181}Hf$ (ref. 21), and the neutron density produced by the still uncertain $^{22}Ne(\alpha, n)^{25}Mg$ reaction.

As described above, all of the calculations so far are based on the assumption that the ratios under consideration are well represented by the steady-state equation (2) and its associated distribution uncertainties for $\tau/\delta > 2$. Still, we need to consider the possibility that $\delta$ may instead be longer than $\tau$. For example, if $\tau/\delta < 0.3$, then it is statistically more likely that the radioactive abundances we observe in the Solar System are exclusively because of the contribution of the last event that enriched the galactic ISM parcel from which the Sun was born[54]. This is the case for the radioactive nuclei $^{129}I$ and $^{247}Cm$ of r-process origin, for which $\delta$ values are larger than their mean lives given the rarity of their stellar sources[22]. In the case of the s-process nuclei, $\delta$ larger than 30–70 Myr would imply an origin from a single event. For $^{107}Pd$ and $^{182}Hf$, it was possible to identify some AGB models that could provide a self-consistent solution, with the best-fit event occurring roughly 25 Myr before the formation of the first solids in the early Solar System[18]. Here we test whether this scenario could also account for the $^{205}Pb/^{204}Pb$ ratios. When considering all three isotopes using the set of Monash models with $Z = 0.014$, stellar masses below roughly 3 $M_\odot$ are not hot enough to produce as much $^{205}Pb$ as needed, whereas models above this mass typically produce too much $^{205}Pb$ and $^{182}Hf$, relative to $^{107}Pd$. The model of mass 3 $M_\odot$ produces self-consistent times around 30 Myr from the last event when using $K = 5.7$ and the lowest $2\sigma$ value of the early Solar System $^{205}Pb/^{204}Pb$ ratio. Overall, a last-event solution may require more fine-tuning than the steady-state solution because, in this case, we do not have any galactic, stochastic uncertainty to allow for a spread in the derived time intervals (as in each panel of Extended Data Fig. 4). Also for this scenario, stellar and nuclear uncertainties need to be carefully evaluated, together with the further constraints that can be derived from the ratios of the radioactive isotopes relative to each other, such as $^{107}Pd/^{182}Hf$ and $^{182}Hf/^{205}Pb$ (refs. 18,134).

Finally, the abundances of all the isotopes considered here may have been contributed to by nucleosynthesis occurring in the massive stars that lived in the same molecular cloud in which the Sun formed and ejected these nuclei within a short enough time to pollute their environment before star formation was extinguished. If such contribution was present and substantial, it needs to be added on top of the contribution that we have calculated here from the AGB stars that evolved before the formation of the molecular cloud and contributed to the chemical evolution of the Galaxy. Wolf–Rayet winds from very massive (>40 $M_\odot$), very short-lived (<5 Myr) and very rare stars may produce $^{107}Pd$ and $^{205}Pb$ (refs. 135,136) but not $^{182}Hf$, which requires higher neutron densities than available in those conditions to activate the branching point at the unstable $^{181}Hf$. Such possible partial contribution does not seem to be required, as GCE already provides a self-consistent solution for all three isotopes together. Core-collapse supernovae, instead, can eject all three isotopes. To provide a successful combination with the GCE contribution, at least according to results calculated with the Monash models, it is required that a potential local core-collapse supernova source produced $^{107}Pd$ and $^{182}Hf$ in similar amounts as in AGB stars and $^{205}Pb$ in potentially higher amounts. This may be achieved, although other factors would play a role in the rich nucleosynthetic environment of a core-collapse supernova, for example, $^{135}Cs$ is expected to be strongly overproduced relative to the current observed upper limit[110], and the long-standing problems of overproduction of $^{53}Mn$ and $^{60}Fe$ by a nearby core-collapse supernova would need to be addressed as well.

## Data availability

Intermediate and result data for the measurement of the bound-state $\beta$ decay of $^{205}Tl^{81+}$ have been published in ref. 81. Source data for Fig. 3 will be published in R.M., T.N. & G.M.-P., manuscript in preparation. Source data for Fig. 4 and Extended Data Fig. 4 are provided in Extended Data Table 3. All of the other relevant data that support the findings of this study are available from the corresponding authors on reasonable request.

## Code availability

The computer code used to analyse the half-life fit of the bound-state $\beta$ decay of $^{205}Tl^{81+}$ from the above-mentioned result data has been published in ref. 82. Details of the code used for the computation of the

weak decay rates will be available in R.M., T.N. & G.M.-P., manuscript in preparation. Details on the code used in Monash models can be found in ref. 49, FUNS models in ref. 50 and NuGrid models in ref. 51. The code used to prepare Fig. 4 and Extended Data Fig. 4 is also published in ref. 82.

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

**Acknowledgements** We sincerely thank all the colleagues who worked towards this measurement over the past 40 years. We dedicate this work to Fritz Bosch, Gottfried Münzenberg, Fritz Nolden, Paul Kienle, Roberto Gallino and Gerald J. Wasserburg, who passed away before they could see the results of this endeavour. Many enlightening discussions with K. Takahashi are greatly appreciated. Advice, support and communication with G. Amthauer, B. Boev, V. Cvetković, B. S. Gao, R. Grisenti, S. Hagmann, W. F. Henning, O. Klepper, M. Lestinsky, A. Ozawa, W. Kutschera, M. K. Pavićević, V. Pejović, D. Schneider, T. Suzuki, S. Yu. Torilov, X. L. Tu, P. M. Walker, N. Winckler, P. J. Woods and X. H. Zhou are acknowledged. The authors express their gratitude to the GSI accelerator team for excellent machine performance, in particular, to R. Heß, C. Peschke and J. Roßbach from the GSI beam cooling group and to GSI management for scheduling flexibility during the COVID-19 pandemic. The results presented here are based on experiment G-17-E121, which was performed at the FRS-ESR facilities at the GSI Helmholtzzentrum für Schwerionenforschung, Darmstadt, Germany, in the frame of FAIR Phase-0. This project has received support from the European Research Council (ERC) under the European Union's Horizon 2020 research and innovation programme (grant agreement no. 682841 'ASTRUm'). The research of G.L., I.D. and C.G. is financed by the Canadian Natural Sciences and Engineering Research Council (NSERC) through grant SAPIN-2019-00030. R.S.S. acknowledges support from the Science and Technology Facilities Council (STFC) (grant no. ST/P004008/1). B.S., M.P. and M.L. acknowledge the support of the ERC Consolidator Grant (Hungary) programme (RADIOSTAR, grant agreement no. 724560) and the Lendület Program LP2023-10 of the Hungarian Academy of Sciences. B.S. was supported by the ÚNKP-23-3 – New National Excellence Programme of the Ministry for Culture and Innovation from the source of the National Research, Development and Innovation Fund. M.L. was also supported by the NKFIH excellence grant TKP2021-NKTA-64. R.M. and G.M.-P. acknowledge support by the Deutsche Forschungsgemeinschaft (DFG, German Research Foundation, project ID 279384907 – SFB 1245). R.M. also acknowledges support provided by the Czech Science Foundation (grant no. 23-06439S). C.B., J.G., Y.A.L., G.M.-P., R.R. and T.S. acknowledge support from the State of Hesse within the Research Cluster ELEMENTS (project ID 500/10.006).

C.B. also acknowledges support from the German Federal Ministry of Education and Research, BMBF, under contracts 05P19RGFA1 and 05P21RGFA1. U.B. and M.P. acknowledge the support to NuGrid from JINA-CEE (NSF grant PHY-1430152) and the continuing access to Viper, the University of Hull's High Performance Computing facility. U.B. and M.P. also acknowledge the support from the European Union's Horizon 2020 research and innovation programme (ChETEC-INFRA – project no. 101008324) and the IReNA network supported by US NSF AccelNet (grant no. OISE-1927130). R.G. acknowledges support by the Excellence Cluster ORIGINS from the DFG (Excellence Strategy EXC-2094-390783311). A.K. was supported by the Australian Research Council Centre of Excellence for All Sky Astrophysics in 3 Dimensions (ASTRO 3D), through project number CE170100013. M.P. appreciates the support from the NKFI through K-project 138031 (Hungary). B.M. and Y.Z. thank the ExtreMe Matter Institute (EMMI) at the GSI Helmholtzzentrum für Schwerionenforschung, Darmstadt, Germany. T.Y. acknowledges support from the Sumitomo Foundation, Mitsubishi Foundation and JSPS KAKENHI nos. 26287036, 17H01123 and 23KK0055. U.B. and M.P. are members of the NuGrid Collaboration (https://nugrid.github.io/).

**Author contributions** G.L., R.S.S., R.J.C., M.B., K.B., C.B., T.D., I.D., D.D., T.F., O.F., B.F., H.G., R.G., J.G., C.G., A.G., E.H., P.-M.H., W.K., C.K., N.K., K.L., S.L., Y.A.L., E.M., T.M., C.N., N.P., U.P., S.P., R.R., S.S., C.S., U.S., M.S., T.S., Y.K.T., S.T., L.V., M.W., H.W., T.Y., Y.Z. and J.Z. prepared and conducted (on site or online) the experiment. Owing to the COVID-19 pandemic, only a limited number of people were allowed on site. G.L., R.S.S., R.J.C., I.D., T.F., R.G., J.G., A.G., W.K., C.K., Y.A.L., S.S. and M.T. analysed the data to extract the $^{205}$Tl$^{81+}$ half-life. R.M., G.M.-P., T.N. and K.L. calculated the weak decay rates. I.D. performed the reevaluation of the neutron-capture cross-sections. B.S., U.B., S.C., A.K., T.K., M.L., B.M., M.P., D.V. and A.Y.L. performed the asymptotic giant branch and galactic chemical evolution modelling. G.L., M.L. and Y.A.L. drafted the manuscript, with contributions from R.M., S.C., I.D., T.F., and G.M.-P. All authors have reviewed, discussed, and commented on the results and the manuscript.

**Competing interests** The authors declare no competing interests.

**Additional information**
**Correspondence and requests for materials** should be addressed to Guy Leckenby, Rui Jiu Chen, Yuri A. Litvinov or Maria Lugaro.

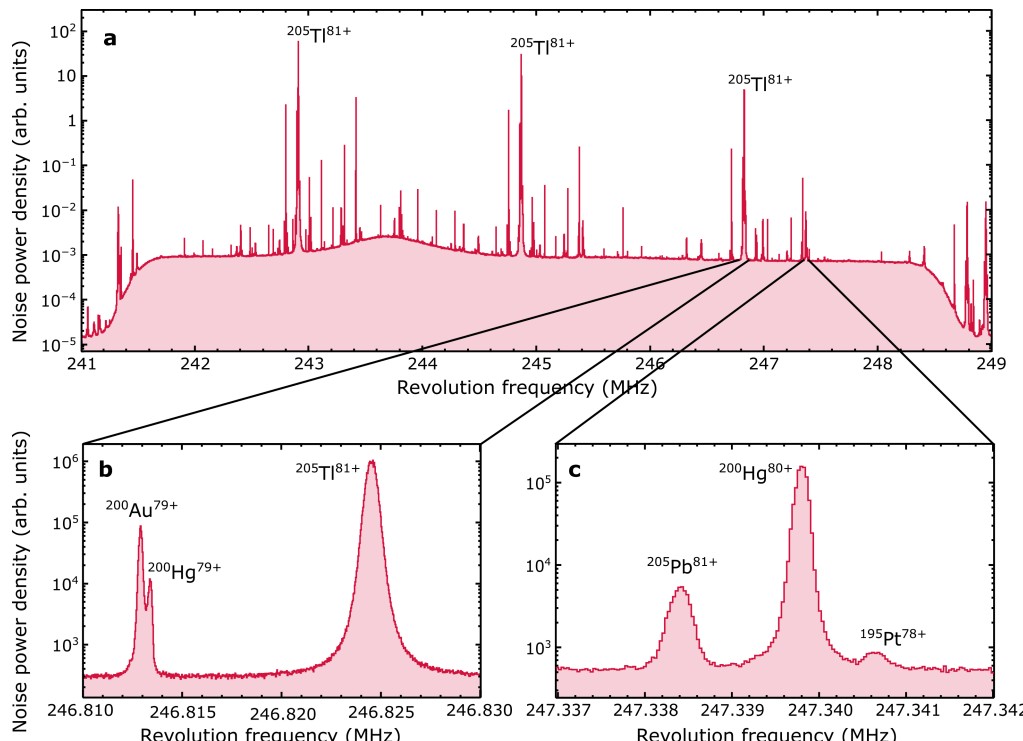

**Extended Data Fig. 1 | Schottky spectrum. a**, The full Schottky spectrum is shown, which includes the 124th, 125th and 126th harmonics. This spectrum was taken during the measurement window. **b**, Zooming in on the $^{205}Tl^{81+}$ peak. **c**, Zooming in on the $^{205}Pb^{82+}$ peak. Both peaks were well separated from the contaminant species.

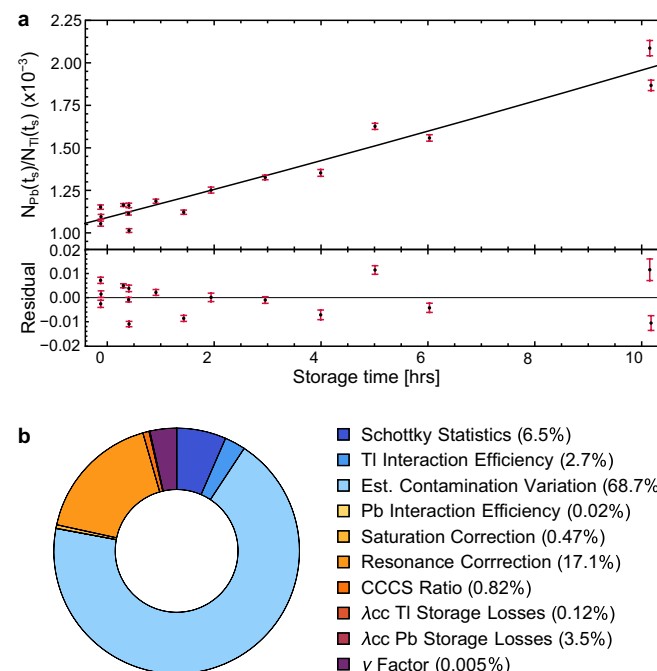

**Extended Data Fig. 2 | Experimental uncertainty details. a**, The raw data with all statistical uncertainty before estimating variance in $^{205}Pb^{81+}$ contamination from projectile fragmentation. The reduced $\chi^2 = 21.6$ demonstrates that we are sensitive to variation in contamination that is unquantified elsewhere in the analysis. **b**, A breakdown of the contributions of each correction to the total uncertainty. Statistical errors are in various blue shades and systematic errors are in various sunset shades.

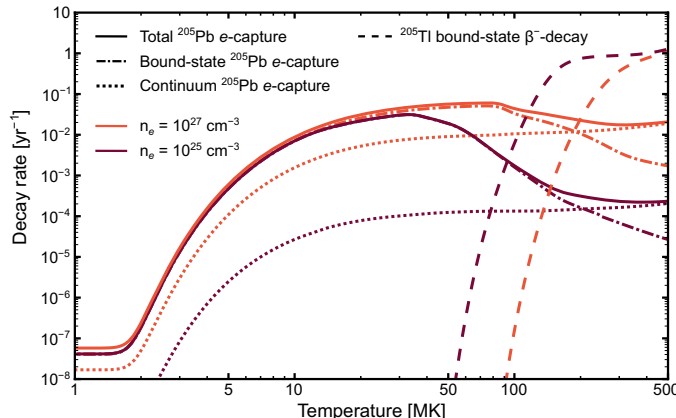

**Extended Data Fig. 3 | Bound and continuum components of weak rates.**
Weak rates connecting $^{205}$Tl and $^{205}$Pb for two different densities as a function
of temperature. For $^{205}$Pb, we show the contributions of bound and continuum
electron capture.

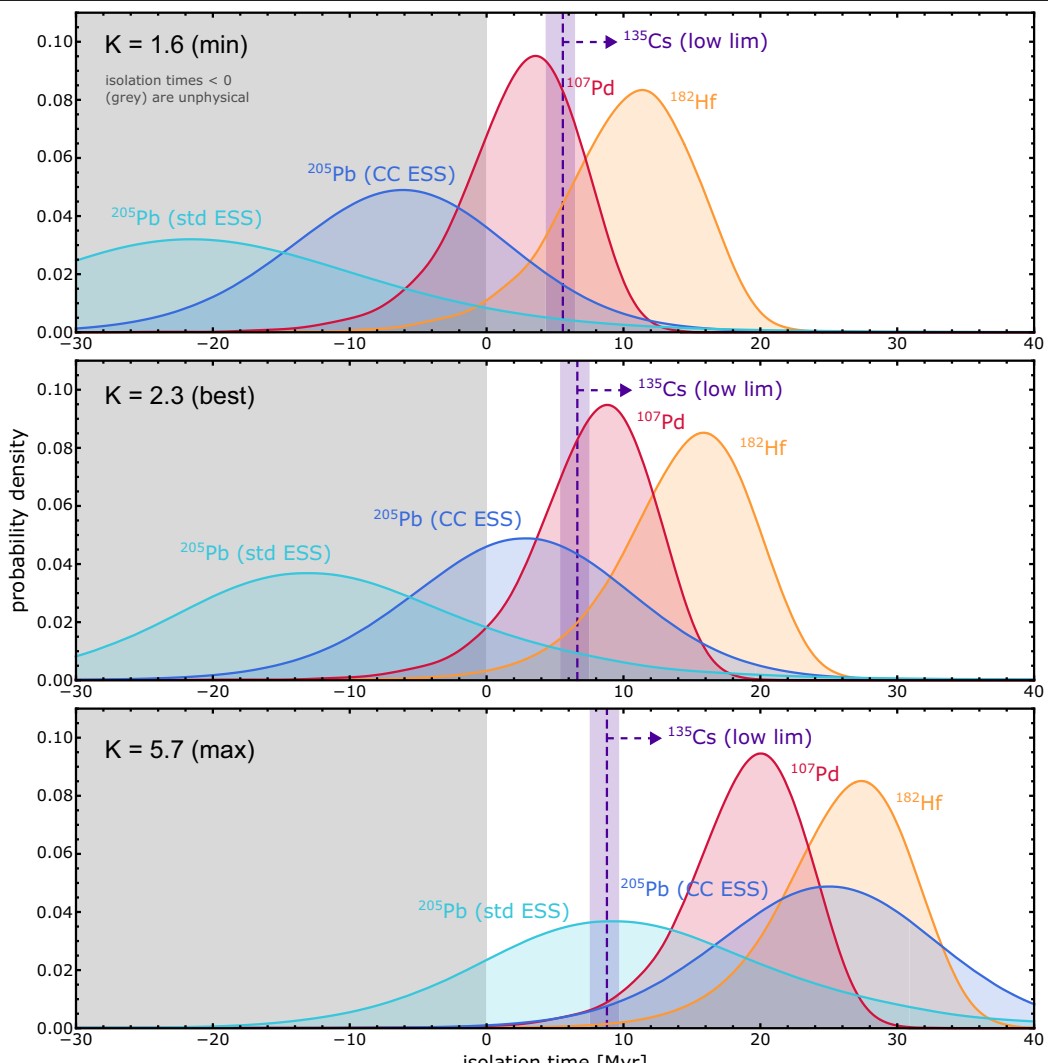

**Extended Data Fig. 4 | Probability density functions for the isolation time with varying K.** The same probability density functions plotted in Fig. 4 are replicated for the min/best/max values of the galactic evolution parameter K following equation (2) and including the lower limit derived from $^{135}$Cs/$^{133}$Cs, which is represented by dashed lines as a reminder that its calculation assumes a different $\delta$ (approximately 1 Myr) than that for the other three isotopes (approximately 3 Myr). ESS and CC represent the standard early Solar System (ESS) value from ref. 9 and the carbonaceous chondrites (CC) ESS value from ref. 10, respectively.

**Extended Data Table 1 | New recommended Maxwellian-averaged cross-sections, SEFs and ground-state contribution factors *X* used for the stellar model calculations**

| | $^{202}$Hg | $^{203}$Hg* | $^{204}$Hg | $^{203}$Tl | $^{204}$Tl | $^{205}$Tl | $^{204}$Pb | $^{205}$Pb* | $^{206}$Pb |
|---|---|---|---|---|---|---|---|---|---|
| kT [keV] | | | | Maxwellian-averaged cross section [mbarn] | | | | | |
| 5 | 200(20) | 1088(713) | 109(51) | 580(18) | 801(179) | 134(34) | 304(15) | 686(153) | 21.3(14) |
| 10 | 124(11) | 644(304) | 63(29) | 305(31) | 497(134) | 86(21) | 166(8) | 446(113) | 19.4(10) |
| 15 | 97(7) | 481(195) | 47(22) | 226(42) | 387(111) | 70(18) | 122(6) | 340(78) | 17.1(9) |
| 20 | 83(5) | 394(150) | 40(19) | 188(46) | 326(101) | 62(16) | 102(5) | 279(56) | 15.6(8) |
| 25 | 74(4) | 339(125) | 37(17) | 166(47) | 286(90) | 56(14) | 91(5) | 240(43) | 14.7(8) |
| 30 | 68(4) | 302(11) | 35(16) | 152(46) | 260(90) | 52(13) | 84(4) | 213(34) | 14.2(9) |
| 40 | 60(3) | 254(92) | 32(15) | 134(42) | 220(79) | 46(11) | 76(4) | 178(23) | 13.5(8) |
| 50 | 55(2) | 224(81) | 30(14) | 123(38) | 198(70) | 41(10) | 71(4) | 156(18) | 12.8(8) |
| 60 | 52(2) | 201 (73) | 29(14) | 115(34) | 176(63) | 38(10) | 68(3) | 140(15) | 12.2(8) |
| 80 | 47(2) | 171(63) | 27(13) | 103(29) | 157(60) | 33(8) | 64(3) | 119(12) | 11.3(8) |
| 100 | 44(2) | 150(57) | 25(12) | 94(26) | 137(64) | 30(7) | 61(3) | 106(10) | 10.7(7) |
| kT [keV] | | | | Stellar enhancement factor | | | | | |
| 5 | 1.000 | 0.987 | 1.000 | 1.000 | 1.000 | 1.000 | 1.000 | 0.972 | 1.000 |
| 10 | 1.000 | 0.963 | 1.000 | 1.000 | 1.000 | 1.000 | 1.000 | 0.956 | 1.000 |
| 15 | 1.000 | 0.939 | 1.000 | 1.000 | 1.000 | 1.000 | 1.000 | 0.946 | 1.000 |
| 20 | 1.000 | 0.914 | 1.000 | 1.000 | 1.000 | 1.000 | 1.000 | 0.939 | 1.000 |
| 25 | 1.000 | 0.890 | 1.000 | 1.000 | 0.998 | 1.000 | 1.000 | 0.933 | 1.000 |
| 30 | 1.000 | 0.869 | 1.000 | 1.000 | 0.995 | 0.999 | 1.000 | 0.927 | 1.000 |
| 40 | 1.000 | 0.835 | 1.000 | 1.000 | 0.985 | 0.996 | 1.000 | 0.919 | 1.000 |
| 50 | 1.000 | 0.810 | 1.000 | 0.999 | 0.970 | 0.990 | 1.000 | 0.911 | 1.000 |
| 60 | 1.000 | 0.792 | 1.000 | 0.997 | 0.952 | 0.981 | 1.000 | 0.903 | 1.000 |
| 80 | 1.000 | 0.774 | 0.997 | 0.992 | 0.915 | 0.960 | 1.000 | 0.886 | 1.000 |
| 100 | 1.002 | 0.776 | 0.991 | 0.989 | 0.886 | 0.942 | 1.000 | 0.869 | 1.000 |
| kT [keV] | | | | Ground-state contribution factor X | | | | | |
| 5 | 1.000 | 0.943 | 1.000 | 1.000 | 1.000 | 1.000 | 1.000 | 0.850 | 1.000 |
| 10 | 1.000 | 0.894 | 1.000 | 1.000 | 1.000 | 1.000 | 1.000 | 0.827 | 1.000 |
| 15 | 1.000 | 0.869 | 1.000 | 1.000 | 1.000 | 1.000 | 1.000 | 0.822 | 1.000 |
| 20 | 1.000 | 0.854 | 1.000 | 1.000 | 1.000 | 1.000 | 1.000 | 0.821 | 1.000 |
| 25 | 1.000 | 0.842 | 1.000 | 1.000 | 0.999 | 1.000 | 1.000 | 0.822 | 1.000 |
| 30 | 1.000 | 0.832 | 1.000 | 1.000 | 0.998 | 0.998 | 1.000 | 0.824 | 1.000 |
| 40 | 1.000 | 0.816 | 1.000 | 0.999 | 0.992 | 0.991 | 1.000 | 0.827 | 1.000 |
| 50 | 0.999 | 0.800 | 0.999 | 0.994 | 0.983 | 0.977 | 1.000 | 0.831 | 1.000 |
| 60 | 0.997 | 0.783 | 0.997 | 0.984 | 0.971 | 0.955 | 1.000 | 0.833 | 1.000 |
| 80 | 0.979 | 0.733 | 0.982 | 0.949 | 0.932 | 0.900 | 1.000 | 0.836 | 1.000 |
| 100 | 0.940 | 0.663 | 0.949 | 0.898 | 0.868 | 0.837 | 0.999 | 0.835 | 0.998 |

All errors are given at 1σ. The form of the uncertainty is discussed in the text and references given therein. SEF and *X* factors are taken from the KADoNiS v1.0 database[48].
*Denotes isotopes for which no experimental information exists.

**Extended Data Table 2 | Predicted mass yields of $^{204,205}$Pb and their ratios from Monash models for AGB stars of solar metallicity $Z=0.014$**

| Mass | $M_{PMZ}$ | $^{205}$Pb | $^{204}$Pb | $^{205}$Pb/$^{204}$Pb |
|------|-----------|------------|------------|-----------------------|
| 2.0 | 0.002 | $1.09 \times 10^{-10}$ | $6.59 \times 10^{-9}$ | 0.017 |
| 2.5 | 0.002 | $5.45 \times 10^{-10}$ | $1.29 \times 10^{-8}$ | 0.042 |
| 3.0 | 0.002 | $1.35 \times 10^{-9}$ | $1.66 \times 10^{-8}$ | 0.081 |
| 3.5 | 0.001 | $2.15 \times 10^{-9}$ | $8.28 \times 10^{-9}$ | 0.259 |
| 4.0 | 0.001 | $6.90 \times 10^{-9}$ | $1.07 \times 10^{-8}$ | 0.645 |
| 4.5 | 0.0001 | $1.38 \times 10^{-9}$ | $2.97 \times 10^{-9}$ | 0.466 |

Masses are in solar units ($M_\odot$). $M_{PMZ}$ is the free parameter in the models that leads to the formation of a 'pocket' rich in $^{13}$C, the main neutron source by means of $^{13}$C($\alpha$,$n$)$^{16}$O. The absolute yields typically increase with the initial stellar mass, unless $M_{PMZ}$ is decreased. The $M_{PMZ}$ trend chosen here is suggested by both models and observations (see ref. 49 for discussion). It does not affect the $^{205}$Pb/$^{204}$Pb ratios, which are mostly a function of the temperature.

**Extended Data Table 3 | ISM ratios calculated (using equation (2)) and isolation times for s-process short-lived radionuclides using the Monash models**

| | $^{205}$Pb/$^{204}$Pb | $^{107}$Pd/$^{108}$Pd | $^{182}$Hf/$^{180}$Hf | $^{135}$Cs/$^{133}$Cs |
|---|---|---|---|---|
| ESS ratio | $1.8(12) \times 10^{-3}$ | $6.6(4) \times 10^{-5}$ | $1.02(4) \times 10^{-4}$ | $< 2.8 \times 10^{-6}$ |
| $\tau$ | 24.5(13) | 9.4(4) | 12.84(13) | 1.9(3) |
| $\delta$ (adopted) | 3.16 | 3.16 | 3.16 | 1.0 |
| $\tau/\delta$ | 7.8 | 3.0 | 4.06 | 1.9 |
| s-process P | 0.167 | 0.141 | 0.111 | 1.059 |
| Unc. factors | 0.76, 1.27 | 0.62, 1.46 | 0.67, 1.39 | 0.53, 1.58 |
| Value of K | *s*-process fraction of stable reference | | | |
| 1.6 (min) | 1 | 0.37 | 0.88 | 0.14 |
| 2.3 (best) | 1 | 0.45 | 0.86 | 0.17 |
| 5.7 (max) | 1 | 0.60 | 0.85 | 0.21 |
| | ISM ratio | | | |
| 1.6 (min) | $(7.7^{+2.1}_{-1.9}) \times 10^{-4}$ | $(9.0^{+4.2}_{-3.5}) \times 10^{-5}$ | $(2.3^{+0.9}_{-0.8}) \times 10^{-4}$ | $(5.1^{+3.0}_{-2.5}) \times 10^{-5}$ |
| 2.3 (best) | $(1.10^{+0.30}_{-0.27}) \times 10^{-3}$ | $(1.57^{+0.73}_{-0.62}) \times 10^{-4}$ | $(3.3^{+1.3}_{-1.1}) \times 10^{-4}$ | $(8.9^{+5.3}_{-4.3}) \times 10^{-5}$ |
| 5.7 (max) | $(2.7^{+0.8}_{-0.7}) \times 10^{-3}$ | $(5.2^{+2.4}_{-2.0}) \times 10^{-4}$ | $(8.0^{+3.1}_{-2.7}) \times 10^{-4}$ | $(2.7^{+1.6}_{-1.3}) \times 10^{-4}$ |
| | Isolation time [Myr] | | | |
| 1.6 (min) | $-20.7^{+11.5}_{-10.0}$ | $2.9^{+3.6}_{-4.7}$ | $10.6^{+4.3}_{-5.3}$ | $> 5.6^{+0.9}_{-1.3}$ |
| 2.3 (best) | $-11.8^{+11.5}_{-10.0}$ | $8.1^{+3.6}_{-4.7}$ | $14.9^{+4.3}_{-5.3}$ | $> 6.6^{+0.9}_{-1.3}$ |
| 5.7 (max) | $10.5^{+11.5}_{-10.0}$ | $19.3^{+3.6}_{-4.7}$ | $26.4^{+4.3}_{-5.3}$ | $> 8.8^{+0.9}_{-1.3}$ |

Uncertainty factors are 1σ equivalent (that is ±1σ = 68.2%) to represent the probability distributions shown in Extended Data Fig. 4. Choice of δ and uncertainty factors are described in the text based on refs. 18,54. ESS ratios (with 2σ uncertainties) are taken from ref. 1 and τ values from ref. 17. Note that we revised the s-process production ratios P to those from Monash AGB models computed with the most recent nuclear inputs, relative to the yields used in ref. 18. The stable s-process fractions are taken from the Monash GCE models of ref. 18, except for $^{204}$Pb, which is an s-only isotope.