## [Peer Review File · Nature]

Manuscript Title: High-temperature ^{205}Tl decay clarifies ^{205}Pb dating in early Solar System

Reviewer Comments & Author Rebuttals

Reviewer Reports on the Initial Version:

Referee #1 (Remarks to the Author):

Key results: Please summarise what you consider to be the outstanding features of the work.

This paper describes the first measurement of the decay of fully ionised ^{205}Tl into ^{205}Pb . ^{205}Pb is an s-nuclide (formed in the s-process) which undergoes a rather slow decay into ^{205}Tl , due to the rather poor structural overlap between ^{205}Pb and ^{205}Tl . However, a (very) low-lying excited state in ^{205}Pb has a much stronger structural overlap which can undergo decay into the ground state of ^{205}Tl at a much higher rate. On the other hand, ^{205}Tl can undergo a bound-state decay into ^{205}Pb under specific situations. A simple interpretation of this is that there isn't enough energy for the beta particle to escape ^{205}Tl but there is enough to launch it into a bound electronic state. However, this bound electronic state is occupied unless ^{205}Tl is in an almost completely ionised state which means that only $^{205}\text{Tl}^{80/81+}$ ions can undergo this process.

Since the temperature at which the s-process activates is high enough that ^{205}Tl is ionised into these high charge states, this bound-state decay process can be activated during the s-process and so understanding how much ^{205}Pb is produced in the s-process requires understanding of this decay pathway. The motivation for studying nuclei such as ^{205}Pb is in understanding the history of our solar system. By investigating the abundances of nuclides formed in the s- and r-processes we are able to constrain how different astrophysical processes contributed to the matter forming the solar system.

Measuring the decay of fully ionised ^{205}Tl requires keeping it in a storage ring so that some of the ions can decay into ^{205}Pb . After injecting and storing the ions and waiting for the decay products to accumulate, the contents of the storage ring interacted with a gas target so that fully stripped $^{205}\text{Pb}^{82+}$ ions could be identified. By counting these $^{205}\text{Pb}^{82+}$ ions, the decay rate of $^{205}\text{Tl}^{81+}$ could be determined.

With the updated decay rate, the ^{204}Pb and ^{205}Pb production in AGB stars was recomputed and a new $^{204}\text{Pb}/^{205}\text{Pb}$ ratio was determined. This ratio was combined with measured Pb and Tl isotopic ratios from presolar grains to determine the isolation time for the material from before the solar system formed. The new results give isolation times in good agreement with those obtained from other long-lived radioisotopes (^{107}Pb and ^{182}Hf) and provide scope for further investigations of the processes which took place during the formation of the solar system.

In my opinion, this is a well-written paper which reports an important and extremely difficult measurement which provides valuable insights on the formation of our solar system. I support its publication but, as ever, have some queries and concerns which I would appreciate the authors

addressing.

Validity: Does the manuscript have flaws which should prohibit its publication? If so, please provide details.

This exceptionally difficult measurement has been carefully performed. There are some details which could be clarified but there are no obvious flaws which cast doubt on the validity of the results presented in the manuscript.

Originality and significance: If the conclusions are not original, please provide relevant references. On a more subjective note, do you feel that the results presented are of immediate interest to many people in your own discipline, and/or to people from several disciplines?

As the authors make clear in the paper, this observation of the ^{205}Tl bound-state beta decay has been long sought-after. The experimental challenges are immense. Production of ^{205}Tl beams from stable samples is complicated by the chemical toxicity of thallium (see, e.g., A. Christie, *The Pale Horse*, Collins Crime Club, 1961) but GSI was able to produce a ^{205}Tl beam by using knockout from a primary ^{206}Pb beam. The resulting $^{205}\text{Tl}^{81+}$ ions were selected and stored in the GSI storage ring.

I think that this experiment could only have been performed at GSI with the storage ring there, at least at the moment. The lifetime which they have successfully measured has been the object of great effort over a long period (as explained in the paper itself), which is testament to the importance of this measurement for nuclear astrophysics.

Data & methodology: Please comment on the validity of the approach, quality of the data and quality of presentation. Please note that we expect our reviewers to review all data, including any extended data and supplementary information. Is the reporting of data and methodology sufficiently detailed and transparent to enable reproducing the results?

In the "Methods" section with the updated (n, γ) cross sections at s-process energies, more details about the updated cross sections may be helpful. Particularly, since a major focus of this paper is on the possible behaviour of nuclei in a stellar plasma when there are elevated temperatures, giving a clear explanation of whether these new recommended MACS values include a "stellar enhancement factor" in their computation would be helpful.

Appropriate use of statistics and treatment of uncertainties: All error bars should be defined in the corresponding figure legends; please comment if that's not the case. Please include in your report a specific comment on the appropriateness of any statistical tests, and the accuracy of the description of any error bars and probability values.

At some points in the manuscript, values are given with the confidence (?) of the uncertainties [e.g. on line 1065 in which the isochrone is given with a slope of $(1.0 \pm 0.4) \times 10^{-3}$ (at 2σ) and in the rest of that paragraph ending on line 1082] but in other places (Table I, for example), the meaning of the uncertainties is not defined. It may also be useful to define the expected form of the uncertainty since cross sections are expected to be log-normally distributed (see e.g. the discussion in

Longland++ Nuclear Physics A 841 1).

In the section entitled “Estimated contamination variation”: I understand why the methodology was used in order to try to estimate the unknown uncertainty in the initial contamination since there’s clearly more variation than can be explained by the other sources of uncertainty. This method of inflating the systematic uncertainty is similar to other methods e.g. making the reduced chi squared equal to 1 to try to account for unknown systematic uncertainties. This is a more advanced method because it includes the intrinsic variation in the chi squared which would be expected from the data.

It would be interesting to see a few more pieces of information about this. First of all, the Bayesian analysis which supports the interpretation would be interesting to see (there doesn’t appear to be any reason to omit it here of which I am aware). Secondly, there’s a growth factor which is included to account for the initial contamination. Are the results for the chi squared analysis consistent with this assumption? I understand the motivation for the choice and it seems valid but it would be good to know if this choice contributes anything to the uncertainty estimation. I am especially interested in this since Fig 5a seems to show that the agreement gets worse with long storage times but this could be a trick of the eye.

Conclusions: Do you find that the conclusions and data interpretation are robust, valid and reliable?

Yes, the conclusion is certainly appropriate and the interpretation flows naturally from the results of the experiment.

Suggested improvements: Please list additional experiments or data that could help strengthening the work in a revision.

I have divided by suggested improvements into two sections, substantive and grammatical/typographical/otherwise minor.

Substantive suggestions

Line 684: I’m not sure that this is substantive but the manuscript mentions that the $^{205}\text{Pb}^{82+}$ ions were subject to the same several processes as the $^{205}\text{Tl}^{81+}$ and $^{205}\text{Pb}^{81+}$ ions. For my own clarity: there are processes which are moving $^{205}\text{Pb}^{81+}$ to $^{205}\text{Pb}^{82+}$ and the reverse all the time. The $^{205}\text{Pb}^{82+}$ ions referred to here include but are not limited to those which are created when the argon gas stripper is deployed? Furthermore, the modification to this rate when there is the argon gas stripper deployment are the corrections discussed in lines 780-787?

Line 700: It would be interesting to know the collision rate (rather than just “low”) because this has some impact on the confidence that a reader has in the overall analysis. It’s certainly my impression that this is well understood since Fig 5 shows a very clear trend.

Line 726-728: MWPCs were deployed to count produced $q=80+$ ions. I've looked a number of times and I cannot find a discussion in the paper about why this was done. My assumption is that it helps with information about capturing an electron by the beam ions in some manner but I cannot see a discussion of how this information was used in this paper. If I've missed it, please accept my apologies.

Lines 1105-1123: There are a number of rates which may be of importance and the nucleosynthesis code uses e.g. various $^{22}\text{Ne}(\alpha, n)$ rates to investigate this. There are also updated (n, γ) rates. What is the source of the other rates used in the model (if any).

Lines 1151-1170: (Table 2) This table includes the results of the Monash calculations but there is no corresponding table for FUNS/NuGrid. Now, answering my own question: I assume that this was because the comparison at 3 solar masses was to test the astro model dependence and nothing else but is this ratio of different values (FUNS being $\sim \frac{1}{3}$ of Monash and NuGrid being ~ 2.5 of Monash) expected to be the same for other masses? Does that make any difference to the overall conclusion?

Also for Table 2: are there uncertainties in these mass yields?

Figure 7: ^{205}Pb (CC ESS) and (std ESS) are not defined in this caption (but are defined in the caption of Figure 4 which may be sufficient).

Minor suggestions

Line 341: "revealed ^{205}Pb ions in the $82+$ charge state" The phrasing here is that ^{205}Pb ions in the $82+$ charge state are revealed but, from the description in the paper, they're actually $^{205}\text{Pb}81+$ ions which are being stripped into the $82+$ charge state?

Line 293: "2.3-keV excited state"?

Line 556: "self-consistent solution" scenario? Or solution in the sense of the same isolation time comes out from the equations.

Line 698: "Capturing an electron by $^{205}\text{Pb}82+$ " should this be "capture of an electron by $^{205}\text{Pb}82+$ "?

Line 715: "charge-exchange reactions"?

Lines 739-740: "The Schottky detector has a wide dynamic range". In what? I assumed frequency but the following statement about being sensitive to single ions made me wonder about whether it's in charge.

Line 761 (after equation 4 but probably referring to equation 1): $\Delta\lambda_{\text{loss}}$ doesn't appear to be defined in either equation 1 or equation 4. I assume that $\Delta\lambda_{\text{loss}} = \lambda_{\text{Tlloss}} - \lambda_{\text{Pbloss}}$ and reflects the difference in the rates at which both Tl and Pb ions are lost from the storage ring due to charge-exchange reactions resulting in them leaving the $81+$ charge state. Certainly that would agree with

the statement between equations 4.17 and 4.18 in Ref. 75.

Line 903: “we notice that...” Should this be “we note that”?

Line 909-911: is there any particular reason for choosing these electron densities? I am trying to ascertain if they correspond to particular astrophysical situations or are just being used for instructive purposes.

Line 1083: it may be useful to give “the ISM ratio of $^{205}\text{Pb}/^{204}\text{Pb}$ ” or something like that. I think that the meaning is clear but there’s a long discussion of Cd and Pt isotopes in the paragraph two before etc.

Line 1178 and line 1183 and line 1190: Nugrid vs NuGrid vs Nugrid. By majority decision “Nugrid” but I saw a talk today which said NuGrid.

References: Does this manuscript reference previous literature appropriately? If not, what references should be included or excluded?

Yes.

Clarity and context: Is the abstract clear, accessible? Are abstract, introduction and conclusions appropriate?

Yes, though I think that it should be “nuclear-physics limitation” and “experimentally backed” in the abstract. I’m sure that a sub-editor can tell me that I’m wrong.

Inflammatory material: Does the manuscript contain any language that is inappropriate or potentially libelous?

No.

Springer Nature is committed to diversity, equity and inclusion; please raise any concerns that may in your view have an impact on this commitment.

N/A

Please indicate any particular part of the manuscript, data, or analyses that you feel is outside the scope of your expertise, or that you were unable to assess fully.

There are certainly things which I’ve asked which come from my ignorance, notably some details on the operation of the storage ring and the models of galactic chemical evolution.

Please address any other specific question asked by the editor via email.

There were no additional questions asked by the editor via email.

Referee #2 (Remarks to the Author):

The manuscript presents the results of a recent measurement of nuclear decay and uses the obtained information to better constrain the ^{205}Pb abundance resulting from the s-process in AGB stars. The experiment is difficult and groundbreaking and would warrant publication on its own. The combination of the experimental result with a theoretical physics estimate of a decay in a stellar plasma, an astrophysical simulation of the resulting nucleosynthesis in a stellar model, and consequently using that nucleosynthesis in a model of Galactic Chemical Evolution (GCE) to derive the expected abundance of ^{205}Pb in the presolar cloud nicely illustrates the importance of an interdisciplinary approach to solve astrophysical questions. Thus, this huge effort combining expertise in several research areas not only yields a timely result of great interest in astrophysics, helping to constrain models of stellar nucleosynthesis as well as GCE, but also serves as a perfect example of interdisciplinary research, also interesting to a wide readership inside and outside of physics.

I would recommend acceptance of the manuscript for publication after the following details have been clarified either in the main text or in the Methods section. These details affect the estimate of the uncertainties in the high-temperature decay and thus also propagate to the uncertainty in the final ^{205}Pb abundance after GCE.

On lines 873-876 in the Methods section it is mentioned that the matrix elements connecting the $\text{Tl } 1/2+$ and $\text{Pb } 1/2-$ states are independent of the weak process considered. This may be the case but they may not be independent of the actual states involved because the overlap between initial and final states may be different. Therefore the matrix element depends on which excited states are involved. In consequence, the matrix element for decay of Tl to the higher lying excited state of Pb may be different than the matrix element for the decay of the low-lying $1/2-$ state of Pb . It is not clear whether this difference has been considered and what additional uncertainty it introduces to the competition between production of ^{205}Pb and decay of ^{205}Pb .

Secondly, it is rightly mentioned that the decay rate of ^{205}Pb is temperature-dependent because of the increasing population of the 2.3 keV state with rising temperature. Calculating the competition between production of ^{205}Pb from ^{205}Tl and the decay of ^{205}Pb , the temperature-dependent occurrence of internal transitions also has to be considered. States in ^{205}Pb between the one at 2.3 keV and the one populated by the Tl decay may also be populated, either by thermal excitations from the g.s. and the 2.3 KeV state or, more importantly, by de-excitation cascades from the state populated by ^{205}Tl decay. States populated in the cascade may also decay and thus affect the balance between production and decay of ^{205}Pb . The current manuscript does not include any discussion of how this was accounted for and whether the additional uncertainty by unknown transition data was considered in the final uncertainty.

Incidentally, in the manuscript I did not find the excitation energies of the states involved in the decay of ^{205}Tl to ^{205}Pb . They should be given explicitly, at least in the caption of Fig. 1.

Referee #3 (Remarks to the Author):

Key experimental result:

Measurement of decay rate of $^{205}\text{Tl}^{81+}$ to $^{205}\text{Pb}^{81+}$ inside a storage ring, the ESR at GSI.

Method used:

$^{205}\text{Tl}^{81+}$ was produced by a nuclear reaction of ^{206}Pb primary beam accelerated in SIS-18 to 678 MeV/u impinging on ^9Be target. The reaction fragments were separated by the FRS. The main contaminant $^{205}\text{Pb}^{81+}$ was suppressed to 0.1% by using slits before entering the ESR. After accumulation and cooling the ions were stored up to 10 hours and the beam intensity continuously monitored. To detect the $^{205}\text{Pb}^{81+}$ produced from the beta decay of $^{205}\text{Tl}^{81+}$, first separation is applied using argon gas-jet for about 10 minutes to strip the bound electron. The beam was simultaneously cooled by increasing the electron cooler current from 20 mA to 200 mA. The detection was realised using different non-destructive detectors. The Schottky detector allows identification of different m/q ratios and therefore distinguish between $^{205}\text{Tl}^{81+}$ and $^{205}\text{Pb}^{82+}$ after passing through the gas-jet. Many sources of systematic error were studied. The dominant source found was the variation in the amount of the contaminant, meaning the count rate variation of $^{205}\text{Pb}^{81+}$ that was injected together with the $^{205}\text{Tl}^{81+}$ beam in the ring.

Comments:

The experimental method is well described and illustrated with appropriate figures. However, some clarifications are needed and are detailed below:

- A key point in the detection is the use of Schottky cavity to distinguish between $^{205}\text{Tl}^{81+}$ and $^{205}\text{Pb}^{82+}$ that was separated using the gas-jet. However, no Schottky spectrum or data are shown to confirm this separation. There is also the detector saturation issue that is not clear how it was overcome. It would be good to show the spectrum before and after the cross-calibration and saturation correction.
- On page 18, line 790, it is said that only correction 3 was determined individually for each storage time but not explained why correction 1, 2 and 4 were not done individually for each injection.
- The main source of systematic error that the authors tried to estimate is the contamination variation. The authors think that it comes from the fragmentation reaction (line 359-361). In principle, if there is variation in such reaction, it should affect all the fragments. Was the production monitored during the experiment? If yes, were there any variation for other fragments? If not, was this variation observed before in other fragmentation reaction at the FRS?
- The final uncertainty in fig. 2 was increased after estimation of the contamination variation to achieve a better fit to the data. Is it possible that other contributions were underestimated and contamination uncertainty overestimated?

There are 3 measurements around storage time 0 (fig.2). Their spread reflects to a certain extent the variation of the contamination, which is about 10%. Naively, one could propagate this uncertainty for other storage times.

Other minor comments:

Line 274-324: repetition "only facility that can provide stored..."

Line 327: not clear what is "enough orbiting".

Line 672-682: Not clear what is outer orbit, inner orbit and middle orbit means. Can there be a schematic to show these orbits and their purpose?

Line 691: it is not clear what the authors mean by "ions moved to the main beam at $q=81+\dots$ "

Line 740-742: This sentence give the impression that the Schottky cavity spectrum could not be properly exploited.

Line 802: It's not clear what the authors mean by "the precision from the Schottky detector meant that we were sensitive to this variation".

Author Rebuttals to Initial Comments:

We would like to thank all referees for their positive comments and thorough feedback on our manuscript, the document is certainly improved with their input. We respond to their comments below, in-line. We conclude, at the end of this document, by highlighting some additional edits we have made to the manuscript.

Referee #1 (Remarks to the Author):

Key results: *Please summarise what you consider to be the outstanding features of the work.*

This paper describes the first measurement of the decay of fully ionised ^{205}Tl into ^{205}Pb . ^{205}Pb is an s-nuclide (formed in the s-process) which undergoes a rather slow decay into ^{205}Tl , due to the rather poor structural overlap between ^{205}Pb and ^{205}Tl . However, a (very) low-lying excited state in ^{205}Pb has a much stronger structural overlap which can undergo decay into the ground state of ^{205}Tl at a much higher rate. On the other hand, ^{205}Tl can undergo a bound-state decay into ^{205}Pb under specific situations. A simple interpretation of this is that there isn't enough energy for the beta particle to escape ^{205}Tl but there is enough to launch it into a bound electronic state. However, this bound electronic state is occupied unless ^{205}Tl is in an almost completely ionised state which means that only $^{205}\text{Tl}^{80/81+}$ ions can undergo this process.

Since the temperature at which the s-process activates is high enough that ^{205}Tl is ionised into these high charge states, this bound-state decay process can be activated during the s-process and so understanding how much ^{205}Pb is produced in the s-process requires and understanding of this decay pathway. The motivation for studying nuclei such as ^{205}Pb is in understanding the history of our solar system. By investigating the abundances of nuclides formed in the s- and r-processes we are able to constrain how different astrophysical processes contributed to the matter forming the solar system.

Measuring the decay of fully ionised ^{205}Tl requires keeping it in a storage ring so that some of the ions can decay into ^{205}Pb . After injecting and storing the ions and waiting for the decay products to accumulate, the contents of the storage ring interacted with a gas target so that fully stripped $^{205}\text{Pb}^{82+}$ ions could be identified. By counting these $^{205}\text{Pb}^{82+}$ ions, the decay rate of $^{205}\text{Tl}^{81+}$ could be determined.

With the updated decay rate, the ^{204}Pb and ^{205}Pb production in AGB stars was recomputed and a new $^{204}\text{Pb}/^{205}\text{Pb}$ ratio was determined. This ratio was combined with measured Pb and Tl isotopic ratios from presolar grains to determine the isolation time for the material from before the solar system formed. The new results give isolation times in good agreement with those obtained from other long-lived radioisotopes (^{107}Pb and ^{182}Hf) and provide scope for further investigations of the processes which took place during the formation of the solar system.

In my opinion, this is a well-written paper which reports an important and extremely difficult measurement which provides valuable insights on the formation of our solar system. I support its publication but, as ever, have some queries and concerns which I would appreciate the authors addressing.

We thank the reviewer for the overall positive evaluation of our work.

Validity: *Does the manuscript have flaws which should prohibit its publication? If so, please provide details.*

This exceptionally difficult measurement has been carefully performed. There are some details which could be clarified but there are no obvious flaws which cast doubt on the validity of the results presented in the manuscript.

Originality and significance: *If the conclusions are not original, please provide relevant references. On a more subjective note, do you feel that the results presented are of immediate interest to many people in your own discipline, and/or to people from several disciplines?*

As the authors make clear in the paper, this observation of the ^{205}Tl bound-state beta decay has been long sought-after. The experimental challenges are immense. Production of ^{205}Tl beams from stable samples is complicated by the chemical toxicity of thallium (see, e.g., A. Christie, *The Pale Horse*, Collins Crime Club, 1961) but GSI was able to produce a ^{205}Tl beam by using knockout from a primary ^{206}Pb beam. The resulting $^{205}\text{Tl}^{81+}$ ions were selected and stored in the GSI storage ring.

I think that this experiment could only have been performed at GSI with the storage ring there, at least at the moment. The lifetime which they have successfully measured has been the object of great effort over a long period (as explained in the paper itself), which is testament to the importance of this measurement for nuclear astrophysics.

Data & methodology: *Please comment on the validity of the approach, quality of the data and quality of presentation. Please note that we expect our reviewers to review all data, including any extended data and supplementary information. Is the reporting of data and methodology sufficiently detailed and transparent to enable reproducing the results?*

In the “Methods” section with the updated (n, γ) cross sections at s-process energies, more details about the updated cross sections may be helpful. Particularly, since a major focus of this paper is on the possible behaviour of nuclei in a stellar plasma when there are elevated temperatures, giving a clear explanation of whether these new recommended MACS values include a “stellar enhancement factor” in their computation would be helpful.

All the given values are Maxwellian-averaged cross sections as measured in the lab, with the target in the ground-state, as given in the KADoNiS database. Stellar modellers, including the Monash, FUNS, and NuGrid members of this collaboration, multiply these values with the respective SEF that are given in the KADoNiS database. We have added a few sentences of explanations to make this clear on lines 1013-1024 of the new manuscript.

In addition, the use of the “stellar enhancement factor” (SEF) as measure of the impact a measured cross section of the nucleus in the ground-state has come under debate in recent years. Instead, for neutron capture cross sections, an “X factor” has been introduced (T. Rauscher *et al.*, *Astrophys. J.* 738, 143 (2011)) that determines the contribution of the ground-state to the stellar rate at a given stellar temperature. Any X factor much smaller than 1 implies that the measured lab rate cannot constrain the stellar rate (captures from excited states) reliably anymore. These X factors are given—together with the SEFs—in the KADoNiS online database. Unfortunately, the KADoNiS database servers at the University of Frankfurt are not configured properly and do not show any datasets at the moment. Several users have contacted the owners a few weeks ago without success so far. To circumvent

this data access problem we have added two tables (Table II and Table III) in the Methods section with the respective values for comparison and modified the text accordingly.

Appropriate use of statistics and treatment of uncertainties: *All error bars should be defined in the corresponding figure legends; please comment if that's not the case. Please include in your report a specific comment on the appropriateness of any statistical tests, and the accuracy of the description of any error bars and probability values.*

At some points in the manuscript, values are given with the confidence (?) of the uncertainties [e.g. on line 1065 in which the isochrone is given with a slope of $(1.0 \pm 0.4) \times 10^{-3}$ (at 2σ) and in the rest of that paragraph ending on line 1082] but in other places (Table I, for example), the meaning of the uncertainties is not defined. It may also be useful to define the expected form of the uncertainty since cross sections are expected to be log-normally distributed (see e.g. the discussion in Longland++ Nuclear Physics A 841 1).

We thank the reviewer for pointing out the places where we missed explanations on the error bars, one of the challenges of interdisciplinary work is navigating different representations of uncertainty. For example, nuclear physics mostly deals in 1σ error bars, but meteoritics reports at 2σ conventionally. We have reviewed the manuscript and explicitly indicated the employed uncertainty margins where they were missing.

In Table 1, due to the very different situation regarding the cross-section data for the nine isotopes, there is no "default" way to extract uncertainties. Many of these cross-section libraries are based on theoretical extrapolations or interpolations from measured datapoints, or have been derived completely from theory. We give some explanation in the text in the Methods section how we think some very conservative error bars can be assigned. Since we do not want to obscure the complexity of the error bars or give false confidence in our error estimation, especially errors associated with theoretical results, we direct the reader to the associated references.

In the section entitled "Estimated contamination variation": I understand why the methodology was used in order to try to estimate the unknown uncertainty in the initial contamination since there's clearly more variation than can be explained by the other sources of uncertainty. This method of inflating the systematic uncertainty is similar to other methods e.g. making the reduced chi squared equal to 1 to try to account for unknown systematic uncertainties. This is a more advanced method because it includes the intrinsic variation in the chi squared which would be expected from the data.

It would be interesting to see a few more pieces of information about this. First of all, the Bayesian analysis which supports the interpretation would be interesting to see (there doesn't appear to be any reason to omit it here of which I am aware).

The Bayesian analysis is indeed interesting, however, we decided to keep the details brief for two reasons: 1) we wanted to avoid having multiple values for the β_b^- -decay rate to avoid confusion, and 2) the Bayesian analysis was implemented as a check and thus has not received the same level of scrutiny as the Monte Carlo analysis. Instead, we plan to publish a comparison of the employed statistical analyses in a separate upcoming work.

To elaborate on the details, two Bayesian analyses were done. The first was done in Mathematica using a standard Jeffrey's prior and the same likelihood as the Monte Carlo analysis, i.e. including a Gaussian "missing uncertainty" with the R_0 growth factor. This analysis yielded a value of $\lambda_{\beta_b} = 2.78(26) \times 10^{-8} \text{ s}^{-1}$, in almost exact agreement with the

Monte Carlo method. This is not surprising though given the formulation is statistically equivalent, and so was mostly done to check that the Monte Carlo method was performing as expected. However, there remained an unsolved issue that the posterior normalised to 1.06.

The second analysis used a cutting-edge Bayesian package called “Nested_fit”, detailed in refs [80—82]. We first did a fit that does not provide a mechanism for including additional uncertainty if the error bars are underestimated, which resulted in a value of $\lambda_{\beta_b} = 2.61(14) \times 10^{-8} \text{ s}^{-1}$. However, we believe that the uncertainty on this result is too small because the associated reduced χ^2 for these error bars is 21.6, as described in the paper.

We also implemented a fit where the quoted error bars were considered to be lower limits σ_0 , so that the data-point uncertainty became a free parameter σ . We apply a “Conservative method” introduced by Sivia (2006, p. 168) that uses a modified Jeffrey’s prior of the form $P(\sigma|\sigma_0, I) = \sigma_0/\sigma^2$, and this generated a fit result of $\lambda_{\beta_b} = 2.34(21) \times 10^{-8} \text{ s}^{-1}$. Because the likelihood penalty is greatly reduced by allowing σ to inflate, the “pull” of outliers on the distribution is greatly reduced. In particular, in the figure below, we can see that the fit has completely departed from the higher lying 10hr point to better fit the bulk of the data points at low storage times. This reduces the “lever arm” potential of the long storage times, which from a counting statistics perspective have the most ^{205}Pb produced by β_b^- decay.

Figure 1: fit result for modified Jeffrey's prior.

Thus, both Bayesian results are consistent with our original Monte Carlo approach within 1σ error bars. But the intricacies are quite nuanced, and we do not want to distract the reader from the methods we currently use. The behaviour of the Bayesian analyses is something we plan to discuss more deeply in a dedicated publication with the lead by the Paris collaborators, who authored the cited references [80—82].

Secondly, there’s a growth factor which is included to account for the initial contamination. Are the results for the chi squared analysis consistent with this assumption? I understand the motivation for the choice and it seems valid but it would be good to know if this choice contributes anything to the uncertainty estimation. I am especially interested in this since Fig 5a seems to show that the agreement gets worse with long storage times but this could be a trick of the eye.

We deeply appreciate the referee’s interest in the statistical rigour of our method. The least squares sum of a data set follows a χ^2 distribution if the members of the sum are normally distributed with $\mu = 0$ and $\sigma = 1$. The formatting of the sum such that $X_i = (d_i - f)/\sigma_i^2$

converts all data points into unit normal distributions by centring them on the fitted equation (i.e. $\mu = 0$) and normalising by the variance (i.e. $\sigma = 1$). The R_0 growth factor ensures that the variance used to normalise the data point is accurately sized, and thus satisfies the requirement that the least squares data be unit normally distributed. In fact, the R_0 growth factor is *required* to satisfy the χ^2 assumption. We have added a sentence to the manuscript stating this at line 907.

It is important to put the R_0 growth factor into perspective though. For the longest 10hr storage times, the growth is equal to a factor of 1.133. So, whilst it is important to consider, it is ultimately a second order effect. For a mechanical perspective, the effect of the growth factor is to allow the 10hr data points to vary 13% more than they otherwise would, slightly reducing the size of the estimated contamination. It is important to highlight that this is a physical effect, because the R_0 contamination grows with storage time due to the differential storage losses, so its variance also grows by the same factor. Ultimately, the growth factor is a model dependent choice because we believe the variation is coming from the initial contamination.

As to whether the inclusion of the growth factor adds anything to the uncertainty estimation, the result with the growth factor is $\lambda_{\beta_b} = 2.761(280) \times 10^{-8} \text{ s}^{-1}$, and the result without is $\lambda_{\beta_b} = 2.761(288) \times 10^{-8} \text{ s}^{-1}$, so the impact is minimal.

In terms of Fig 5a, one must be careful evaluating the agreement of the data points beyond the fact that the error bars are not large enough to explain the scatter. If one did decide to evaluate whether data points were consistent with these small error bars, there are several points at lower storage times that are much more statistically significant due to their very small error bars. The scatter may appear slightly larger at longer storage times, but this is completely expected. Every systematic uncertainty (except the saturation correction) increases with storage time because we have lower beam intensity at the time of measurement, which is reflected in the larger error bars in Fig 5a. To repeat though, it is obvious that not all the variation is represented by the error bars in Fig 5a, so evaluating agreement is hazardous.

Conclusions: *Do you find that the conclusions and data interpretation are robust, valid and reliable?*

Yes, the conclusion is certainly appropriate and the interpretation flows naturally from the results of the experiment.

Suggested improvements: *Please list additional experiments or data that could help strengthening the work in a revision.*

I have divided by suggested improvements into two sections, substantive and grammatical/typographical/otherwise minor.

Substantive suggestions

- Line 684: I'm not sure that this is substantive but the manuscript mentions that the $^{205}\text{Pb}82+$ ions were subject to the same several processes as the $^{205}\text{Tl}81+$ and $^{205}\text{Pb}81+$ ions. For my own clarity: there are processes which are moving $^{205}\text{Pb}81+$ to $^{205}\text{Pb}82+$ and the reverse all the time. The $^{205}\text{Pb}82+$ ions referred to here include but are not limited to those which are created when the argon gas stripper is deployed? Furthermore, the modification to this rate when there is the argon gas stripper deployment are the corrections discussed in lines 780-787?

This is an insightful observation. Indeed, the ions in the storage ring are subject to a variety of processes, and their populations need to be modelled by a coupled differential equation. The referee's observation that the processes moving $^{205}\text{Pb}^{81+}$ to $^{205}\text{Pb}^{82+}$ and the reverse all the time is correct, but the physics behind these processes changes in different stages of the measurement. We are preparing a manuscript with details on this differential equation, which we fit to certain parts of the Schottky data (made very challenging by the saturation effect) to extract estimates on the rates of these processes. Additionally, we would like to note that these atomic charge-exchange processes have been extensively studied at the ESR since 1990s and are meanwhile very well described theoretically.

During storage, where just electron cooling and rest gas is relevant, the electron recombination rate for $^{205}\text{Pb}^{82+}$ was 30 times faster than the stripping rate for $^{205}\text{Pb}^{81+}$, meaning that most ^{205}Pb remained in the 81+ charge state. During stripping in the gas target however, the opposite was true with electron stripping of $^{205}\text{Pb}^{81+}$ dominating recombination of $^{205}\text{Pb}^{82+}$ by a factor of 30. Processes occurring during the storage time are a factor of 4000 slower than during the gas-target stripping, so we can comfortably assume that gas target stripping is the dominant mechanism beyond the observation that ^{205}Pb ions affected by other processes remain stored. The corrections discussed in lines 832—838 are indeed parameterisations of the dominant gas-target stripping effect. Given the reviewer correctly assessed what was occurring with the current information, the fact we already have a lengthy manuscript, and given a future publication with more detail, we suggest keeping the text as it is.

- Line 700: It would be interesting to know the collision rate (rather than just “low”) because this has some impact on the confidence that a reader has in the overall analysis. It's certainly my impression that this is well understood since Fig 5 shows a very clear trend.
Indeed, the beam lifetime is well understood. Our conclusion on the “low” collision rate stems from the fact that the storage times of several hours were achieved at beam velocities of about 70% of speed of light. Note that beam losses induced by recombination in the electron cooler and recombination caused by collisions with the residual gas are not separated because they result in the same atomic loss channel. Recombination in the rest gas depends critically on the gas composition and especially on the amount of heavy gas atoms/molecules. Although theoretical modelling for every gas species is very accurate, the composition uncertainty propagates into the calculations. The standard assumption for the ESR rest gas after a long pumping period (which is valid here) is 80% of H_2 and 20% of N_2 . However, a tiny pollution by Ar from the jet target cannot be excluded, which was however not identified in the rest gas analysis. Therefore, it is difficult to provide an exact number on the collision rate. However, beam lifetimes are straightforwardly accurately obtained experimentally to a great accuracy employing non-destructive current measurement, e.g., with Schottky detectors. We have added a sentence providing the beam lifetime (lines 709-714 in the new manuscript).
- Line 726-728: MWPCs were deployed to count produced $q=80+$ ions. I've looked a number of times and I cannot find a discussion in the paper about why this was done. My assumption is that it helps with information about capturing an electron by the beam ions in some manner but I cannot see a discussion of how this information was used in this paper. If I've missed it, please accept my apologies.
The MWPC detectors were deployed during gas-target stripping as a complementary measurement of the beam lifetime and were necessary in determining the charge-

changing cross section ratio. The beam lifetime measurement proved particularly fortuitous due to the issues we had with the Schottky data acquisition as it allowed us to extract very accurate beam lifetimes (see attached thesis section 3.4.5 for more details). We have added this sentence to line 738 in the new manuscript for clarity.

- Lines 1105-1123: There are a number of rates which may be of importance and the nucleosynthesis code uses e.g. various $^{22}\text{Ne}(\alpha, n)$ rates to investigate this. There are also updated (n, γ) rates. What is the source of the other rates used in the model (if any).

As the referee has identified, the choice of nuclear rates is very important. In particular, state-of-the-art AGB models use over 2000 reactions in a coupled reaction network. The choices of reactions are discussed in the works cited when introducing each AGB model, as the choices vary based on the model. We do not want to overwhelm the reader with details that are not pertinent to our result, especially given the large amount of moving pieces we already have, and so have not directly mentioned the source of less important rates. As the referee identifies, the $^{22}\text{Ne}(\alpha, n)$ rate is very important as the neutron source that powers the s-process which contributes active ^{205}Pb , and so we discuss our choices on lines 1225-1227 (new manuscript), noting that other choices of $^{22}\text{Ne}(\alpha, n)$ makes less than a 10% difference. Similarly for the updated (n, γ) rates, which are new evaluations specifically for this work.

For the referee's interest, other rates used in the Monash models include:

- Most of (p, γ) reactions are either from the reaclib fit of Raucher & Thielemann, 2000 or Cyburt et al. 2010.
- Triple- α and $^{12}\text{C}+\alpha$ from Fynbo et al. 2005
- $^{13}\text{C}(\alpha, n)$ from Heil et al 2008
- Most neutron-capture rates came from JINA reaclib, but for 24 isotopes, the rates are directly from the KaDoNiS database (v0.2) The Zr neutron capture cross sections are from Lugaro et al 2014 ApJ.
- Decay rates from NETGEN except ^{181}Hf decay from Lugaro et al 2014 science and ^{134}Cs decay from Li, ... Lugaro... et al 2021.

The rates used by FUNS and NuGrid models are of course different.

- Lines 1151-1170: (Table 2) This table includes the results of the Monash calculations but there is no corresponding table for FUNS/NuGrid. Now, answering my own question: I assume that this was because the comparison at 3 solar masses was to test the astro model dependence and nothing else but is this ratio of different values (FUNS being $\sim\frac{1}{3}$ of Monash and NuGrid being ~ 2.5 of Monash) expected to be the same for other masses? Does that make any difference to the overall conclusion? The referee is correct in assessing that the $3 M_{\odot}$, $Z = 0.02$ star was used to test the model dependence; this was our test mass and metallicity as we refined our method. To provide another point of comparison, we have also run a $2 M_{\odot}$, $Z = 0.02$ model for FUNS and NuGrid, which yielded factors of 0.30 and 3.57 respectively when compared to the $2 M_{\odot}$, $Z = 0.014$ Monash model. Thus, we can see that whilst other mass models would probably vary slightly due to the complexities involved, given that all models describe the same underlying physics, we expect to see similar trends across masses. In particular, the temperature in the intershell, which is the most predictive factor for ^{205}Pb production, increases with mass in a fairly consistent way. We suggest not including this additional check in the manuscript, however, in part because the metallicity is not precisely equal (unlike for the $3 M_{\odot}$ case), but also because $3 M_{\odot}$ AGB stars are more representative of the population that would dominate ^{205}Pb production due to their higher yields.

Given the scope of potential work involved in calculating tables for FUNS and NuGrid models, we believe that such calculations are beyond the scope of this paper. Mass-dependent yields are not the only avenue for extension of this work, we would also like to consider metallicity-dependent yields by using a full numerical GCE code, as was done by Trueman et al. 2022. We have already noted that metallicity dependent yields also have an impact on the final result (see lines 1321-1327 of the new manuscript), so we hope to do a comprehensive comparison of models with a more sophisticated treatment of GCE than the steady-state assumption of equation (2).

With all this in mind, we concluded that reproducing the analysis done for Monash models for both FUNS and NuGrid models would involve a lot of additional work for little added scientific impact in the context of this paper, but the colleagues working on individual models are inspired to address exactly this in future works.

- Also for Table 2: are there uncertainties in these mass yields?

Whilst we appreciate and wholeheartedly agree with the referee's desire to have uncertainties on the mass yields, this is simply not possible because there are too many uncertainties that cannot be meaningfully quantified in a single stellar evolution + nucleosynthesis code. On top of the nuclear reactions, the mass loss and the mixing at convective borders are the main uncertainties, also affected by processes related to rotation and magnetic fields. There is a whole industry of stellar modellers who try to tackle each of these problems. Furthermore, while some uncertainties can be parameterized (still by a variety of functions and sets of free parameters), others, such as mixing are also intrinsic to the numerical methods employed by each code. For example, the Monash results are based on a version of the Monash stellar evolution code that does not include e.g. diffusion and extends borders of convective regions beyond the Schwarzschild criterion only if there is a discontinuity in the temperature gradient, and by adding one radiative mass shell to the convective zone. The post-processing Monash code takes this structure input and treats mixing region using an advecting scheme for convective regions, which is most suitable for nucleosynthesis. Therefore, the best way to evaluate such systematic uncertainties, at least qualitatively, is to compare results from different models, as we have done in the paper. We believe that choosing three very different codes gives a good, conservative estimate of the stellar uncertainties as the results from all these codes are well studied and have been compared to observations before. A full discussion of the treatment of uncertain physics is well beyond the scope of the paper.

- Figure 7: ^{205}Pb (CC ESS) and (std ESS) are not defined in this caption (but are defined in the caption of Figure 4 which may be sufficient).

We thank the referee for pointing out this oversight and have included a definition in the figure caption (line 1369 in new manuscript).

Minor suggestions

- Line 341: "revealed ^{205}Pb ions in the 82+ charge state" The phrasing here is that ^{205}Pb ions in the 82+ charge state are revealed but, from the description in the paper, they're actually $^{205}\text{Pb}^{81+}$ ions which are being stripped into the 82+ charge state? We have rephrased this sentence to be more straightforward, see lines 339/340.
- Line 293: "2.3-keV excited state"?
We are not sure if the line number is correct here, but we have hyphenated all examples of 2.3-keV excited state.

- Line 556: “self-consistent solution” scenario? Or solution in the sense of the same isolation time comes out from the equations.
Solution was intended to mean solution to the problem of explaining the origin of the short-lived radionuclides. We have replaced solution with scenario to be more clear.
- Line 698: “Capturing an electron by 205Pb82+” should this be “capture of an electron by 205Pb82+”?
Indeed, the suggested change sounds better, we have implemented it.
- Line 715: “charge-exchange reactions”?
Implemented.
- Lines 739-740: “The Schottky detector has a wide dynamic range”. In what? I assumed frequency but the following statement about being sensitive to single ions made me wonder about whether it’s in charge.
The large dynamic range refers to the amplitude of the ion induced signals, meaning that the Schottky detector itself is sensitive to very low as well as very high excitation amplitudes without any distortion, even in the same spectrum. Distortion can only be caused by the signal processing elements that follow, such as low noise amplifiers, power amplifiers, RF switches and most importantly the data acquisition system. Such distortions are known in the field of communications, and come in a variety of forms, like the "Third-order intercept point" or just clipping to rail amplitude.
To clarify this for the reader, we have included part of this explanation in the text on line 750-753.
- Line 761 (after equation 4 but probably referring to equation 1): $\Delta\lambda_{loss}$ doesn’t appear to be defined in either equation 1 or equation 4. I assume that $\Delta\lambda_{loss} = \lambda_{TI_{loss}} - \lambda_{Pb_{loss}}$ and reflects the difference in the rates at which both TI and Pb ions are lost from the storage ring due to charge-exchange reactions resulting in them leaving the 81+ charge state. Certainly that would agree with the statement between equations 4.17 and 4.18 in Ref. 75.
We thank the referee for noting the lack of definition, and note that the assumption is correct. Since we don’t use the term $\Delta\lambda^{loss}$ again, we have replaced it with $\lambda_{TI}^{loss} - \lambda_{Pb}^{loss}$.
- Line 903: “we notice that...” Should this be “we note that”?
Implemented.
- Line 909-911: is there any particular reason for choosing these electron densities? I am trying to ascertain if they correspond to particular astrophysical situations or are just being used for instructive purposes.
They are mostly for illustrative purposes. The range of electron densities we have presented are relevant for AGB stars, with 10^{24} being relevant for the surface of the star and 10^{28} being relevant for the base of the intershell during the thermal pulse. The specific values of 10^{25} and 10^{27} were chosen as good examples of the different contributions of bound and continuum electron capture.
In reviewing Fig 6, we also noted that the labels of bound and continuum electron capture had been swapped, this has also been fixed.
- Line 1083: it may be useful to give “the ISM ratio of 205Pb/204Pb” or something like that. I think that the meaning is clear but there’s a long discussion of Cd and Pt isotopes in the paragraph two before etc.
Implemented: the sentence now reads “the $^{205}\text{Pb}/^{204}\text{Pb}$ ISM ratio of”.
- Line 1178 and line 1183 and line 1190: Nugrid vs NuGrid vs Nugrid. By majority decision “Nugrid” but I saw a talk today which said NuGrid.
We agree with the referee that NuGrid is correct and thank them for pointing out this oversight.

References: *Does this manuscript reference previous literature appropriately? If not, what references should be included or excluded?*

Yes.

Clarity and context: *Is the abstract clear, accessible? Are abstract, introduction and conclusions appropriate?*

Yes, though I think that it should be “nuclear-physics limitation” and “experimentally backed” in the abstract. I’m sure that a sub-editor can tell me that I’m wrong.

We have implemented these suggestions, but also defer to the copy-editor.

Referee #2 (Remarks to the Author):

The manuscript presents the results of a recent measurement of nuclear decay and uses the obtained information to better constrain the ^{205}Pb abundance resulting from the s-process in AGB stars. The experiment is difficult and groundbreaking and would warrant publication on its own. The combination of the experimental result with a theoretical physics estimate of a decay in a stellar plasma, an astrophysical simulation of the resulting nucleosynthesis in a stellar model, and consequently using that nucleosynthesis in a model of Galactic Chemical Evolution (GCE) to derive the expected abundance of ^{205}Pb in the presolar cloud nicely illustrates the importance of an interdisciplinary approach to solve astrophysical questions. Thus, this huge effort combining expertise in several research areas not only yields a timely result of great interest in astrophysics, helping to constrain models of stellar nucleosynthesis as well as GCE, but also serves as a perfect example of interdisciplinary research, also interesting to a wide readership inside and outside of physics.

I would recommend acceptance of the manuscript for publication after the following details have been clarified either in the main text or in the Methods section. These details affect the estimate of the uncertainties in the high-temperature decay and thus also propagate to the uncertainty in the final ^{205}Pb abundance after GCE.

We thank the referee for their kind words. In addition, we would like to apologise for what we believe may have been a misleading representation of the nuclear structure that we presented in Fig. 1b. In the original submission, we plotted the energetics of the decay of atomic ^{205}Pb on the bottom and of bare ions of ^{205}Tl on top using the same energy scale. The energy scale was intended to compare the differences in atomic binding energies of the two situations to help readers understand why the decay path inverts for H-like and bare ions. However, we now see that this representation could have been confusing. To rectify this, we have changed Fig. 1b to have independent energy scales for the neutral and ionised cases.

To be 100% clear, the $1/2 +$ state in $^{205}\text{Tl}^{81+}$ and the $5/2 -$ and $1/2 -$ states in $^{205}\text{Pb}^{81+}$ are the same ground and first excited nuclear states as in neutral ^{205}Tl and ^{205}Pb .

On lines 873-876 in the Methods section it is mentioned that the matrix elements connecting the Tl $1/2+$ and Pb $1/2-$ states are independent of the weak process considered. This may be the case but they may not be independent of the actual states involved because the overlap between initial and final states may be different. Therefore the matrix element depends on which excited states are involved. In consequence, the matrix element for decay of Tl to the higher lying excited state of Pb may be different than the matrix element for the decay of the low-lying $1/2-$ state of Pb. It is not clear whether this difference has been considered and what additional uncertainty it introduces to the competition between production of ^{205}Pb and decay of ^{205}Pb .

In the original version of Fig 1b, we intended to show the energetics for the electron capture of atomic ^{205}Pb on the bottom and that for the bound decay of bare ^{205}Tl ions on top. The nuclear states shown on top and bottom ($5/2 -$ ground state and the $1/2 -$ first excited state in ^{205}Pb and the $1/2 +$ ground state in ^{205}Tl) are the same states and therefore also the nuclear matrix elements involved in the electron capture of ^{205}Pb and the bound decay of ^{205}Tl are the same. What differs between the levels on the top and on the bottom are the electronic configurations.

Secondly, it is rightly mentioned that the decay rate of ^{205}Pb is temperature-dependent because of the increasing population of the 2.3 keV state with rising temperature.

Calculating the competition between production of ^{205}Pb from ^{205}Tl and the decay of ^{205}Pb , the temperature-dependent occurrence of internal transitions also has to be considered. States in ^{205}Pb between the one at 2.3 keV and the one populated by the Tl decay may also be populated, either by thermal excitations from the g.s. and the 2.3 KeV state or, more importantly, by de-excitation cascades from the state populated by ^{205}Tl decay. States populated in the cascade may also decay and thus affect the balance between production and decay of ^{205}Pb . The current manuscript does not include any discussion of how this was accounted for and whether the additional uncertainty by unknown transition data was considered in the final uncertainty.

As discussed above the levels on the top part of Fig. 1b reflect the bound decay of bare ^{205}Tl ions. The states populated by the ^{205}Tl decay are the first excited state in ^{205}Pb at 2.3 keV or, with a very small probability, the ground state, both in $^{205}\text{Pb}^{81+}$ ions with one electron in the K-shell. These are the only states that can be populated in the decay of the $1/2 +$ state in $^{205}\text{Tl}^{81+}$. Other transitions only become possible with the thermal population of the next excited states (a $3/2 +$ state at 204 keV in ^{205}Tl and a $3/2 -$ state at 263 keV in ^{205}Pb) at very high temperatures. These transitions from the higher lying excited states are included in our rate calculations but are not discussed in the paper as they only provide a very minor contribution.

Incidentally, in the manuscript I did not find the excitation energies of the states involved in the decay of ^{205}Tl to ^{205}Pb . They should be given explicitly, at least in the caption of Fig. 1. As explained above, these states are the ground and first excited state at 2.3 keV in $^{205}\text{Pb}^{81+}$ ions.

Referee #3 (Remarks to the Author):

Key experimental result:

Measurement of decay rate of $^{205}\text{Tl}^{81+}$ to $^{205}\text{Pb}^{81+}$ inside a storage ring, the ESR at GSI.

Method used:

$^{205}\text{Tl}^{81+}$ was produced by a nuclear reaction of ^{206}Pb primary beam accelerated in SIS-18 to 678 MeV/u impinging on ^9Be target. The reaction fragments were separated by the FRS. The main contaminant $^{205}\text{Pb}^{81+}$ was suppressed to 0.1% by using slits before entering the ESR. After accumulation and cooling the ions were stored up to 10 hours and the beam intensity continuously monitored. To detect the $^{205}\text{Pb}^{81+}$ produced from the beta decay of $^{205}\text{Tl}^{81+}$, first separation is applied using argon gas-jet for about 10 minutes to strip the bound electron. The beam was simultaneously cooled by increasing the electron cooler current from 20 mA to 200 mA. The detection was realised using different non-destructive detectors. The Schottky detector allows identification of different m/q ratios and therefore distinguish between $^{205}\text{Tl}^{81+}$ and $^{205}\text{Pb}^{82+}$ after passing through the gas-jet. Many sources of systematic error were studied. The dominant source found was the variation in the amount of the contaminant, meaning the count rate variation of $^{205}\text{Pb}^{81+}$ that was injected together with the $^{205}\text{Tl}^{81+}$ beam in the ring.

Comments:

The experimental method is well described and illustrated with appropriate figures. However, some clarifications are needed and are detailed below:

- A key point in the detection is the use of Schottky cavity to distinguish between $^{205}\text{Tl}^{81+}$ and $^{205}\text{Pb}^{82+}$ that was separated using the gas-jet. However, no Schottky spectrum or data are shown to confirm this separation. There is also the detector saturation issue that is not clear how it was overcome. It would be good to show the spectrum before and after the cross-calibration and saturation correction.
We thank the referee for pointing this out and have added a figure with the Schottky spectra after gas stripping is complete. This figure now shows how well separated the relevant peaks are.
We believe that explaining how the saturation issue was solved would be too much detail for even the Methods section of this paper. This is why we direct readers to refs. 78,79 for further details. Whilst these references describe in detail the methods used, we acknowledge that they are no longer the most recent description, and therefore we have added a comment directing readers to the upcoming thesis of G. Leckenby, who describes the final methodology. Because this thesis is in the process of being defended, we attach the relevant chapter for the referee's reference.
- On page 18, line 790, it is said that only correction 3 was determined individually for each storage time but not explained why correction 1, 2 and 4 were not done individually for each injection.
The choice to correct individually or globally was based on whether the correction could have plausibly varied with each measurement, or whether it was physical property that was the same for each measurement. For example, correction 3, the gas

stripping efficiency, varied with the gas target density, and we were able to clearly see this correlation (see figure below) so we corrected each measurement individually.

Figure 2: Correlation between the interaction decay constant vs gas target density

In comparison, correction 2, the resonance correction, is a property of the resonance response of the Schottky detector. The variation in revolution frequency between measurements, which is controlled by the stability of the ESR magnets, was negligibly small (tens of Hz vs 500 kHz, see below) to account for the scatter in values we extracted for the resonance correction, and so we attributed the scatter to uncertainty in resonance correction determination rather than real variation in the resonance correction value.

To make this more clear to the reader, we have added a sentence for each correction briefly explaining how each correction was determined and whether it was applied individually or globally.

- The main source of systematic error that the authors tried to estimate is the contamination variation. The authors think that it comes from the fragmentation reaction (line 359-361). In principle, if there is variation in such reaction, it should affect all the fragments. Was the production monitored during the experiment? If yes, were there any variation for other fragments? If not, was this variation observed before in other fragmentation reaction at the FRS?

We thank the referee for addressing this important issue. The variation in contamination is not due to the reaction product but mainly due to instabilities of magnetic fields. The FRS magnetic fields have temporal stability on the order of 10^{-6} . With the slit configuration and optical setting of the FRS, we ensured that only a small fraction of the $^{205}\text{Pb}^{81+}$ distribution was transmitted to the ESR (roughly a $> 3.6\sigma$ tail). However, full separation of TI and Pb ions is impossible (at least at present facilities). Since the intensity of $^{205}\text{Pb}^{81+}$ from the target is huge, this instability is sufficient to cause observed fluctuations in the transmission of $^{205}\text{Pb}^{81+}$. We verified this hypothesis with dedicated Monte-Carlo transport code MOCADI which is used to design experiments at the FRS-ESR.

Transmission fluctuations are different for different fragments depending on their specific trajectory through the ion-optical system of the FRS-ESR. Thus, fluctuations in other contaminants do not necessarily provide constraints on fluctuations in the

$^{205}\text{Pb}^{81+}$ tails. We think that we have found a valid and robust approach to account for transmission fluctuations.

- The final uncertainty in fig. 2 was increased after estimation of the contamination variation to achieve a better fit to the data. Is it possible that other contributions were underestimated and contamination uncertainty overestimated?

As the referee is hinting at, it is true that when you use the scatter of the data to estimate missing uncertainty, you are sensitive to everything that hasn't been accounted for already. In our data analysis, we considered this method of estimating the contamination variation as last resort and have left no stone unturned in searching for other sources of uncertainty. We are confident that with the data we have, all other sources of uncertainty are determined as accurately as possible. We encourage the referee to look at the attached thesis chapter for more details.

The main reason we are confident that the other sources of uncertainty do not encroach on the estimation of the contamination variation is that most of our corrections cannot produce variation in the data, but rather would move all data points up or down together if the true value was slightly different. This leaves only the Schottky statistics and the TI correction efficiency, both of which are very well under control. Thus, we believe the other uncertainty contributions are estimated well.

- There are 3 measurements around storage time 0 (fig.2). Their spread reflects to a certain extent the variation of the contamination, which is about 10%. Naively, one could propagate this uncertainty for other storage times.

This observation is spot on, and it is an important check to ensure everything makes sense. In fact, this was how we initially did our analysis, in that we used the storage time 0 measurements to determine the contamination and then did a contamination subtraction. However, given the direct production of $^{205}\text{Pb}^{81+}$ in the fragmentation reaction dominates our signal for most storage times, we wanted to use every data point together to constrain the contamination variation. Thus, by considering the χ^2 of the data, we can extract the same spread information using the residuals and have 16 data points rather than just 3. When considering a reduced χ^2 of 1, the size of the missing uncertainty is 6%, which is consistent with the observed scatter in the first 3 measurements. In conclusion, we maintain that our Monte Carlo sampling of the χ^2 gives a more robust, but still consistent, method of estimating the contamination variation than just considering for $t_s = 0$ measurements.

Other minor comments:

- Line 274-324: repetition “only facility that can provide stored...”
We have replaced the first instance with “Heavy-ion storage rings are uniquely capable...”
- Line 327: not clear what is “enough orbiting”.
We have replaced “enough” with “over 10^6 ”.
- Line 672-682: Not clear what is outer orbit, inner orbit and middle orbit means. Can there be a schematic to show these orbits and their purpose?
The storage ring has aperture size of 250 mm in diameter in straight sections and flat wide chambers in dispersive places. We refer to the figure below. Here plotted is the beginning of the cycle. The time runs upwards. At zero seconds, an injection of a fresh beam from the FRS occurs on the outer orbit (circumference is about 108.4 m), characterized by a revolution frequency of ~ 246.6 MHz (horizontal axis). We note that 126th harmonic of the revolution frequency is shown. At this orbit, stochastic precooling is acting. Once finished, the beam is captured by an rf bucket and transported to inside of the ring. The orbit length is about 108.2 m and the corresponding larger frequency is

247.3 MHz. Here the electron cooling keeps the stack together. At 15 s and later at 40 s new injections are repeated. The injection is specially designed to inject the new ions on the outside orbits that the ions stored in the inside are unaffected. These are the accumulation steps. At 45 s the accumulated stack is moved to the central orbit by ramping the magnets of the ESR. The central orbit is at 108.3 m. The frequency of 246.8 MHz cannot be directly compared to the previous two values since the magnetic rigidity of the ring is altered at this stage. For more details please see the labels on the figure.

Regarding an earlier comment on the correction 2, one can see that the frequencies of the ions are very stable on this scale, which shall be compared with the resonance response of the Schottky detector (FWHM) of about 1 MHz.

These steps including the figure below are explained in great detail in the PhD thesis of Ragandeep Singh Sidhu, ref. [?] which can be downloaded here:

<https://archiv.ub.uni-heidelberg.de/volltextserver/30275/>.

We have added a sentence on line 673/674 to clarify what is meant by outer vs inner orbits. To avoid extending an already long manuscript, we direct the reader to the above thesis rather than including the below figure.

Figure 7.8: Different stacked ions are shown in the inner and outer orbit of the ESR for HN 126.

- Line 691: it is not clear what the authors mean by “ions moved to the main beam at $q=81+\dots$ ”

We have restructured the sentence to read “Similar electron recombination for $^{205}\text{Pb}^{82+}$ ions reduced their charge state to $q = 81 +$, where they returned to the main beam and remained in the ESR.”

- Line 740-742: This sentence gives the impression that the Schottky cavity spectrum could not be properly exploited.
Unfortunately, this is the correct impression. In many aspects of the experiment the Schottky spectrum was too saturated to be used and the current comparator and other auxiliary detectors had to be relied upon. In particular, various decay constants that would normally be measured with the Schottky cavities had to be determined by other means.
- Line 802: It's not clear what the authors mean by "the precision from the Schottky detector meant that we were sensitive to this variation".
The intention of this sentence was to comment on the fact that we were surprised that the other sources of uncertainty could be controlled well enough such that the contamination variation was the dominant form of uncertainty. As mentioned above, in previous FRS experiments, such variations had not been a limiting factor.
Since this sentiment does not provide much insight scientifically, we have replaced it with a more meaningful sentence (lines 855-857 of the new manuscript): "However, by cutting away everything but the extreme tails of the $^{205}\text{Pb}^{81+}$ fragmentation distribution, the impact of instabilities in the yield becomes significant."

Additional Edits

In addition to the edits requested by the above referees, we also draw the attention of both the editor and referees to some other improvements we have made to the resubmitted manuscript:

- **Data and Code Publication:** on lines 1521-1535, we have updated the data and code availability statements to reflect that we have published intermediate + result data and the half-life analysis code to Zenodo. Currently the files are protected, we will release these files to the public at publication. If the editor or referees are interested in the files, the access links are: Data publication link and Code publication link.
- **Update to Monash model outputs:** during the review period, M. Lugaro and B. Szányi identified a bug in the Monash models that slightly altered the mass yields. This affected all yields in a similar, model-dependent way, and so the relevant $^{205}\text{Pb}/^{204}\text{Pb}$ ratios are essentially unchanged. As a result, Fig 4, Fig 8, and Table 4 have been updated with the corrected values. Our scientific conclusions remain unchanged.
- **Updated comparison to GCE study:** during the review period, we revisited the comparison of equation (2) to the full GCE study by Trueman *et al.* 2022 [18]. We found that by controlling the legacy values of the production ratios, we got a more accurate comparison that demonstrated equation (2) performed better than we previously stated. We updated the description of the comparison on lines 1324-1327.
- **Line 249:** "internal transition" was replaced with "internal conversion" to be more specific and the relevant reference was added.
- **Line 369:** we have added a reference to the other crucial motivating case for ^{205}Tl , the determination of the neutrino capture cross section for measuring Solar pp neutrinos in the LOREX project. These results will be reported in a different publication.
- **Line 1547:** we have included some additional names in the acknowledgements section.

Reviewer Reports on the First Revision:

Referee #1 (Remarks on code availability):

I reviewed the PDF which seems to describe the step-by-step analysis process and this seemed sensible. I do not have Mathematica and so I could not run the notebooks provided. The README doesn't provide particular instructions for use but I assume that a Mathematica user would know how to run the notebooks so this doesn't seem like a particular problem to me.

Referee #2 (Remarks to the Author):

The authors have taken great care in their very detailed answers to the referees' questions and have succeeded in clarifying their approach. They also considerably improved content and accessibility in the revised manuscript. I strongly recommend acceptance for publication.

Referee #3 (Remarks to the Author):

The authors addressed all the points raised in the first report. The PhD thesis they sent is very helpful since it details all the technical parts. Presumably, this thesis will be public, which will give more credibility to this publication.

In conclusion, the key experimental results are well described and appropriately referenced. The data and methodology are validated through robust statistical methods, uncertainties are well-addressed, and conclusions are reliable.

For the above reasons I recommend this manuscript for publication in Nature.

Author Rebuttals to First Revision:

Referee #1 (Remarks to the Author):

I thank the authors for their clear and comprehensive responses to my questions, which have all been addressed. I agree with many of their points, especially on stellar enhancement factors and the various details as to the stellar models. I am particularly encouraged that they plan follow-up publications on e.g. expanding the Monash analysis to other models and the Bayesian analysis and I look forward to reading those papers.

All of my queries have been completely addressed and I strongly support publication of this paper.

Referee #1 (Remarks on code availability):

I reviewed the PDF which seems to describe the step-by-step analysis process and this seemed sensible. I do not have Mathematica and so I could not run the notebooks provided. The README doesn't provide particular instructions for use but I assume that a Mathematica user would know how to run the notebooks so this doesn't seem like a particular problem to me.

As the referee correctly guessed, Mathematica code is cell based just like Jupyter notebooks, so the notebook should compile without any specific instructions.

Referee #2 (Remarks to the Author):

The authors have taken great care in their very detailed answers to the referees' questions and have succeeded in clarifying their approach. They also considerably improved content and accessibility in the revised manuscript. I strongly recommend acceptance for publication.

Referee #3 (Remarks to the Author):

The authors addressed all the points raised in the first report. The PhD thesis they sent is very helpful since it details all the technical parts. Presumably, this thesis will be public, which will give more credibility to this publication.

The PhD thesis will be available publicly sometime in December 2024.

In conclusion, the key experimental results are well described and appropriately referenced. The data and methodology are validated through robust statistical methods, uncertainties are well-addressed, and conclusions are reliable.

For the above reasons I recommend this manuscript for publication in Nature.